# OUT-OF-DISTRIBUTION DETECTION IN CLASS INCREMENTAL LEARNING

## ABSTRACT

Class incremental learning (CIL) aims to learn a model that can not only incrementally accommodate new classes, but also maintain the learned knowledge of old classes. Out-of-distribution (OOD) detection in CIL is to retain this incremental learning ability, while being able to reject unknown samples that are drawn from different distributions of the learned classes. This capability is crucial to the safety of deploying CIL models in open worlds. However, despite remarkable advancements in the respective CIL and OOD detection, there lacks a systematic and large-scale benchmark to assess the capability of advanced CIL models in detecting OOD samples. To fill this gap, in this study we design a comprehensive empirical study to establish such a benchmark, named **OpenCIL**, offering a unified protocol for enabling CIL models with different OOD detectors using two principled OOD detection frameworks. One key observation we find through our comprehensive evaluation is that the CIL models can be severely biased towards the OOD samples and newly added classes when they are exposed to open environments. Motivated by this, we further propose a novel approach for OOD detection in CIL, namely Bi-directional Energy Regularization (**BER**), which is specially designed to mitigate these two biases in different CIL models by having energy regularization on both old and new classes. Extensive experiments show that BER can substantially improve the OOD detection capability across a range of CIL models, achieving state-of-the-art performance on the OpenCIL benchmark.

## 1 INTRODUCTION

Training of deep neural networks (DNNs) heavily relies on large-scale data on a fixed set of classes (Russakovsky et al., 2015; Krizhevsky et al., 2017), but data in real-world applications is constantly changing, leading to continuous new classes in training data. Continual learning (CL) enables DNNs to continuously learn a sequence of tasks, with each task consisting of a set of unique classes. The samples for the learned/old tasks are assumed to be not accessible in such dynamic environments. Class incremental learning (CIL) is one type of CL where task identifiers are not known at testing time. Compared to another type of CL, task-incremental learning, CIL is often considered a more practical setting in real applications (Rebuffi et al., 2017; Li & Hoiem, 2017). Thus, we focus on CIL in this study.

Many CIL methods have been introduced over the years to overcome *Catastrophic Forgetting* (CF) of the knowledge learned on old classes, *i.e.*, degraded classification accuracy on old classes due to model updating on new classes (Xiao et al., 2023; Wang et al., 2022a; Niu et al., 2024). They have shown remarkable performance on the in-distribution (ID) classes in incremental tasks, but lack the capability to recognize and reject out-of-distribution (OOD) samples that are drawn from non-training datasets during incremental learning (Huang & Li, 2021; Wang et al., 2020) (*i.e.*, no class overlapping between ID and OOD samples). Such a capability is crucial to the safety of deploying CIL models in open environments in real-world application systems such as autonomous systems (Kendall & Gal, 2017; Leibig et al., 2017). For example, in Unmanned Aerial Vehicles (UAVs), the DNNs are initially trained using a limited set of available trajectories and then continuously updated based on newly available trajectories. Meanwhile, these DNNs need to be capable of recognizing OOD data to handle unexpected situations in every ongoing trajectory.

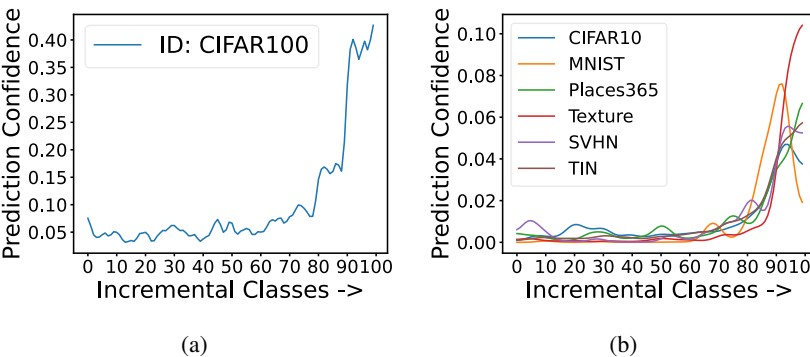

Figure 1: Results of the CIL model iCaRL (Rebuffi et al., 2017) on CIFAR100 (Krizhevsky et al., 2009). **(a)** Mean prediction confidence of iCaRL on test samples from all incremental classes. **(b)** Mean prediction confidence of iCaRL classifying six OOD datasets into one of the ID classes based on the final incremental task. (see Appendix D for the results of other CIL models)

Numerous OOD detection methods have been proposed (Sun et al., 2021; Wang et al., 2023b; Li et al., 2024b; Miao et al., 2024), but these OOD methods are focused on static environments, where the samples of all tasks are accessible during training, making them ineffective in dynamic CIL environments. Therefore, it is non-trivial to combine the off-the-shelf CIL and OOD models. To justify this difficulty, a plausible evaluation protocol is needed to assess the OOD detection capability of different CIL models with the support of different types of OOD detectors. There are such protocols on the respective CIL and OOD detection areas, *e.g.*, OpenOOD (Yang et al., 2022; Zhang et al., 2023) for OOD detection and FACIL (Masana et al., 2022) for CIL, but no work has been done on a systematic and large-scale benchmarking study to evaluate the synergy of existing state-of-the-art (SOTA) CIL models and OOD detection methods.

To bridge this gap, we design a performance benchmark for OOD detection in CIL, called **Open-CIL**, offering a unified protocol for different CIL models with diverse OOD detectors. To achieve this, OpenCIL introduces two principled frameworks for incorporating diverse OOD detection methods into CIL models and also introduces a new evaluation pipeline that enables fair comparison of not only the OOD detection capability for different CIL models but also the ability of different OOD detectors in the presence of CF. In particular, OpenCIL accommodates four representative CIL models with 15 diverse OOD detection methods, resulting in 60 baseline models on two popular CIL datasets and six commonly-used near/far OOD datasets. Based on the large-scale experiments on OpenCIL, we provide a number of important observations, offering crucial insights into the design of CIL models for open-world applications.

One key observation we find is that compared to OOD detection in static environments, the dynamic environments in the CIL setting can lead to increasing biases towards OOD samples and newly added classes with the growth of incremental learning steps. The underlying reasons are two-fold. One main reason is that due to the CF problem, CIL models often have lower prediction confidence for samples of old ID classes (*i.e.*, classes seen in the old tasks), compared to new ID classes, as illustrated in Fig. 1a. This leads to a difficulty in distinguishing between old ID class samples and OOD samples. Furthermore, as illustrated in Fig. 1b, DNN-based CIL models typically exhibit an over-confident prediction on OOD samples, causing the misclassification of the OOD samples into not only the old ID classes but also the new ID classes. Increasing the incremental learning steps results in a larger classification semantic space and more severe CF, which continually amplifies the biases towards the OOD samples and newly added classes. Motivated by these issues, we propose a new approach for OOD detection in CIL, namely Bi-directional Energy Regularization (**BER**), which jointly optimizes two energy regularization terms to modulate the energy prediction of OOD samples w.r.t. samples of old and new class samples, respectively. This effectively reduces these two biases, improving the OOD detection capability across a range of CIL models.

In summary, our main contributions are as follows:

- We investigate the synergy between CIL and OOD detection models, and establish the first benchmark, OpenCIL, for evaluating the OOD detection capability of CIL models and promoting the development of more advanced methods for this under-explored problem.
- We further introduce BER, a novel approach that provides an effective framework for mitigating increasing biases of CIL models towards OOD samples and newly added classes with the growth of incremental steps. This helps largely improve the OOD detection capability of a wide range of CIL models.
- Extensive experiments show that BER achieves state-of-the-art performance on the OpenCIL benchmark under varying incremental step sizes on popular CIL and OOD datasets.

## 2 RELATED WORK

**Out-of-distribution (OOD) Detection.** The objective of this task is to determine whether a given input sample belongs to the learned classes (in-distribution) or unknown classes (out-of-distribution). In recent years, OOD detection has been extensively developed, including Post-hoc-based methods (Sun et al., 2021; Wang et al., 2023b; Zhang & Xiang, 2023) and fine-tuning-based methods (Liu et al., 2020; Wei et al., 2022; Tian et al., 2022; Yu et al., 2023; Li et al., 2023; Liu et al., 2023b; Miao et al., 2024; Li et al., 2024b;a). The post-hoc methods focus on devising new OOD scoring functions in the inference stage. The fine-tuning-based methods focus on separating OOD samples from ID samples by training a strong classifier as OOD detector. However, all these methods are applied to non-CIL models. There lacks of exploration of their capability on CIL models, resulting in poor performance when there is catastrophic forgetting. Our BER is a fine-tuning method that can be applied to different CIL models to improve their OOD detection performance.

**Class Incremental Learning (CIL).** CIL performs the learning procedure in an incremental manner with growing data samples. It focuses on alleviating the catastrophic forgetting problem, in which the CIL models are required to remember the knowledge of the learned classes from old tasks while learning the discriminative information for the newly coming classes. There are three main lines of work in this area (Luo et al., 2023; Xiao et al., 2023). Regularization-based methods focus on applying discrepancy (between old and new models) as penalization terms in their objective functions (Liu et al., 2021; Rebuffi et al., 2017; Xiao et al., 2023). Parameter-isolation-based methods aim to increase the model parameters in each new incremental step to prevent knowledge forgetting caused by parameter overwritten when learning new tasks (Xu & Zhu, 2018; Yan et al., 2021; Wang et al., 2022a). Replay-based algorithms assume there is a memory budget allowing a handful of old class examples in the memory. These memory examples can be used to re-train/fine-tune the CIL model in each new incremental step (Rebuffi et al., 2017; Wu et al., 2019; Luo et al., 2023; Niu et al., 2024). However, all these methods focus on tackling the CIL problem in a closed world, failing to take into account distinguishing ID data from unknown samples (*e.g.*, OOD data), Our BER baselines can be applied to different pre-trained CIL models, in which all new class data and old class memory are used for fine-tuning a new OOD detector for the CIL models. It effectively equips the CIL models with significantly improved capability for OOD detection.

## 3 EQUIPPING CIL MODELS WITH OOD DETECTION CAPABILITY

This section investigates the effectiveness of two principled OOD detection frameworks in enabling CIL models to reject unknown samples.

**Problem Statement.** Our goal is to equip CIL models with the capability of rejecting OOD samples. In this setting, a model is required to learn a sequence of tasks during training. At testing time, the model is used to classify samples into old/new classes while also rejecting unknown samples at each incremental step. The model is evaluated based on its effectiveness in preventing the CF problem and distinguishing samples of old/new classes from OOD data.

Formally, CIL models are learned from a sequence of $c$ tasks ID data $T = \{T_1, T_2, ..., T_c\}$. For each $t$-th ($1 \leq t \leq c$) task, we have $T_t = (X_t^{train}, X_t^{test}, Y_t)$, where $X_t^{train}$ denotes the training ID data, $X_t^{test}$ denotes the testing ID data, and $Y_t$ denotes the classification semantic space consisting of a set of unique classes, *i.e.*, the label spaces between any two incremental tasks have no class

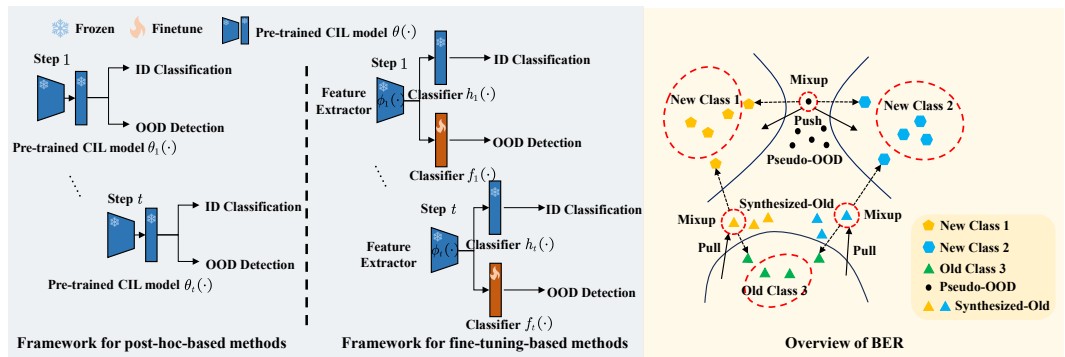

Figure 2: **Left:** Two principled frameworks are used in OpenCIL to incorporate OOD detection methods into the different CIL models. Both frameworks are performed on pre-trained CIL models $\theta(\cdot)$, *i.e.*, $\theta(\cdot)$ is a composition of $\phi(\cdot)$ and $h(\cdot)$, which keep frozen throughout, ensuring that their CIL classification performance is not affected. Post-hoc-based OOD methods are directly applied to pre-trained CIL models. Fine-tuning-based OOD methods train an additional classifier $f(\cdot)$, and apply OOD detection based on this new classifier. **Right:** Our proposed BER aims to leverage two sample synthesis methods to better train $f(\cdot)$, including interpolating samples of new classes to synthesize the pseudo OOD samples for enlarging their decision boundary margin to mitigate bias towards new classes, and interpolating samples of new and old classes to synthesize enhanced old samples for expanding old class decision boundary to mitigate bias towards OOD samples.

overlapping: when $i \neq j$, $Y_i \cap Y_j = \emptyset$. Thus, the set of all seen classes at task $t$ can be denoted as $Q_t = \cup_{i=1}^{t} Y_i$. CIL assumes that data samples of the old classes are not accessible. Many CIL methods, such as the replay-based methods, assume the availability of a memory buffer, in which a memory block is assigned to each task to store a very small set of samples for the task, *i.e.*, $B_i \subseteq (X_i^{train}, Y_i)$ is a small subset of training ID data sampled from task $T_i$. Thus, we have the memory $M_t = \cup_{i=1}^{t-1} B_i$ that includes replay data from all $t$ tasks. Let $\theta_t(\cdot)$ be the CIL model at the incremental learning step $t$, then the memory and the training data of task $t$ form the training ID data: $T_t^{train} = (X_t^{train}, Y_t) \cup M_t$ for training $\theta_t(\cdot)$ at the learning step $t$. For replay-free CIL methods, $\theta_t(\cdot)$ is trained with $(X_t^{train}, Y_t)$ only. During testing time at task $t$, let $T_t^{test} = X_1^{test} \cup X_2^{test} \cup ... \cup X_t^{test}$ be the testing ID data from all $t$ tasks, $\theta_t(\cdot)$ is used to classify samples in $T_t^{test}$ into one of the classes in $Q_t$. This is the setting for the conventional CIL problem.

For OOD detection in CIL, in addition to $T_t^{test}$, the CIL model $\theta_t(\cdot)$ is also presented with an OOD dataset $X_t^{ood}$ at task $t$, which is a set of samples of unknown classes drawn from a different distribution as the ID data in $T_t$. Then given test data $x \in T_t^{test} \cup X_t^{ood}$, the goal of the CIL model $\theta_t(\cdot)$ is to either classify $x$ into the correct ID class from old/new tasks, or detect it as OOD data.

### 3.1 OpenCIL Benchmark: Enabling CIL Models with Existing OOD Detectors

It is challenging for CIL models to recognize OOD samples and to conduct a fair comparison of OOD detection capabilities, given the diverse types of OOD detectors, the significant differences of two settings, and the various incremental steps involved. Therefore, we introduce **OpenCIL**, the first large-scale and systematic benchmark designed to enable CIL models with existing OOD detectors. There are two types of OOD detection methods: post-hoc-based and fine-tuning-based methods. Post-hoc methods calculate OOD scores based on features/logits derived from the pre-trained model, which can be used for different pre-trained classification models. Fine-tuning methods require the fine-tuning of part or all layers of the pre-trained models, and then calculating the OOD score based on the fine-tuned models. Below we introduce two principled frameworks that OpenCIL uses to incorporate these two types of OOD methods into CIL models, as also illustrated in Fig. 2 **Left**.

**CIL models with post-hoc-based OOD detection methods.** A SOTA CIL algorithm is first applied to learn the $c$-tasks stream of ID data in an incremental manner, resulting in the standard CIL model $\theta(\cdot) = \{\theta_1(\cdot), \theta_2(\cdot), ..., \theta_c(\cdot)\}$. Subsequently, we directly perform the post-hoc OOD scoring function on the features/logits extracted from the well-trained CIL model $\theta_t(\cdot), t \in \{1, 2, ..., c\}$ at each incremental step to calculate the OOD score, without having any effect on the CIL models.

**CIL models with fine-tuning-based OOD detection methods.** Similar to the previous framework, fine-tuning-based OOD methods also apply a SOTA CIL algorithm to obtain the standard CIL model $\theta(\cdot) = \{\theta_1(\cdot), \theta_2(\cdot), ..., \theta_c(\cdot)\}$ in the first place. As shown in Fig. 2, for each incremental step, the well-trained CIL model $\theta_t(\cdot), t \in \{1, 2, ..., c\}$ contains a feature extractor $\phi_t(\cdot)$ and a classifier $h_t(\cdot)$, *i.e.*, $\theta_t(\cdot)$ is a composition of $\phi_t(\cdot)$ and $h_t(\cdot)$. Then we freeze both $\phi_t(\cdot)$ and $h_t(\cdot)$, and fine-tune an additional classifier $f_t(\cdot)$ only on top of $\phi_t(\cdot)$ to avoid intensifying the catastrophic forgetting problem. This means that the fine-tuned CIL model contains the same feature extractor $\phi_t(\cdot)$ and classifier $h_t(\cdot)$ as the standard pre-trained CIL model for incremental ID classification, but it also has the additional fine-tuned classifier $f_t(\cdot)$ for OOD detection at each incremental step $t$, without affecting any of the learning procedures of the CIL model $\theta_t(\cdot)$ (*i.e.*, $\phi_t(\cdot)$ and $h_t(\cdot)$). Notably, the fine-tuning of $f_t(\cdot)$ is applied to each incremental step only with training ID data $T_t^{train}$.

With these two principled frameworks, different OOD detection methods can be easily incorporated into three different types of CIL models without impairing the incremental learning accuracy at all. The overall algorithm of these two principled frameworks is provided in Appendix F.

### 3.1.1 BENCHMARK SETUP

**CIL Datasets.** Following (Wang et al., 2022a; Wu et al., 2019; Wang et al., 2023a; Luo et al., 2023), we use two popular CIL datasets as the ID data in our benchmark: CIFAR100 (Krizhevsky et al., 2009) and large-scale ImageNet1k (Russakovsky et al., 2015). Besides, these two datasets are also widely used as ID datasets in the area of OOD detection. Following (Rebuffi et al., 2017; Wang et al., 2022a; Wu et al., 2019), the splits of the two ID datasets are as follows. 1) For **CIFAR100**, we train the CIL model gradually with $k$ classes per incremental step with a fixed memory size of 2,000 exemplars. We respectively evaluate the performance with the step size $k \in \{5, 10, 20\}$. 2) For **ImageNet1K**, since it is a much larger dataset, we train the CIL model gradually with a larger number of $k$ classes per step that $k \in \{50, 100, 200\}$ with a fixed memory size of 20,000 exemplars.

**OOD Datasets.** Following the recent large-scale solely OOD detection benchmark OpenOOD (Yang et al., 2022; Zhang et al., 2023), we select six datasets as OOD data for each ID dataset respectively. 1) For the ID dataset **CIFAR100**, the OOD data includes two near OOD datasets – CIFAR10 (Krizhevsky et al., 2009) and Tiny-ImageNet (TIN) (Le & Yang, 2015) – and four far OOD datasets: MNIST (LeCun et al., 2010), Texture (Cimpoi et al., 2014), SVHN (Netzer et al., 2011) and Places365 (Zhou et al., 2017). 2) For **ImageNet1k**, the OOD data includes four near OOD datasets – ImageNet_O (Hendrycks et al., 2021), iNaturalist (Van Horn et al., 2018), OpenImage_O (Wang et al., 2022b) and Species (Basart et al., 2022) – and two far OOD datasets: MNIST (LeCun et al., 2010) and Texture (Cimpoi et al., 2014).

It is notable that, although these datasets have been widely used in the CIL and OOD detection community respectively, there is still no readily accessible and unified protocol for combining them into one experimental setting. Our OpenCIL benchmark offers one way to unify them into a systematic evaluation setting, facilitating the application of diverse OOD detectors under different CIL models.

**Evaluation Metrics.** To fairly compare the OOD detection performance among different incremental steps, we keep the ratio of testing OOD data $X_t^{ood}$ to testing ID data $T_t^{test}$ fixed at each incremental step $t$. Specifically, we control the test OOD data fed into the CIL model in an incremental manner during the inference stage, increasing the number of testing OOD data $X_t^{ood}$ with increasing number of tasks proportionally. Formally, the number of OOD samples for each OOD dataset at step $t$ is defined as:

$$|X_t^{ood}| = |X^{ood}| \times \frac{t}{|T|}, \tag{1}$$

where $|X^{ood}|$ and $|X_t^{ood}|$ denote the number of OOD samples in the full OOD dataset and in the testing OOD data used at step $t$ respectively, and $T$ is the total number of CIL tasks/steps. In this way, the number of testing ID and OOD data grows at the same speed, keeping the ratio of them identical across all incremental steps, supporting the fair comparison of OOD detection performance among different incremental steps. Following *average incremental accuracy* (Rebuffi et al., 2017), a popular metric in the standard CIL evaluation that assesses the average classification accuracy across all incremental steps, we report the average OOD performance across all incremental steps.

More details about the datasets used and the performance metrics are provided in Appendix A.

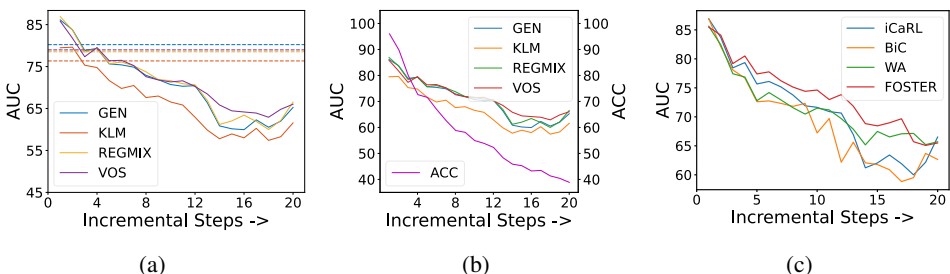

(a)                                     (b)                                     (c)

Figure 3: **(a)** All four representative OOD detection methods experience a decreased AUC performance with increasing incremental steps, compared to themselves working on the full training data of all steps. **(b)** ACC performance decreases quickly throughout all incremental steps, while the AUC performance of all four OOD methods decreases slowly and then levels off. Both (a) and (b) are average performance on six OOD datasets at each incremental step, where the CIL model iCaRL (Rebuffi et al., 2017) is used. ACC is the accuracy of iCaRL on CIFAR100 at each step. The results for the other three CIL models are provided in Appendix D. **(c)** Average performance of CIL models with the OOD detector REGMIX (Pinto et al., 2023) on six OOD datasets at each incremental step on CIFAR100. The results for the other OOD methods are provided in Appendix D.

### 3.1.2 BASELINE LIBRARY

15 OOD detection methods are used in OpenCIL, including nine post-hoc methods – MSP (Hendrycks & Gimpel, 2017), ODIN (Liang et al., 2018), Energy (Liu et al., 2020), MaxLogit (Basart et al., 2022), GEN (Liu et al., 2023a), ReAct (Sun et al., 2021), KLM (Basart et al., 2022), Relation (Kim et al., 2023), and NNGuide (Park et al., 2023) and six fine-tuning-based methods – LogitNorm (Wei et al., 2022), T2FNorm (Regmi et al., 2023), AUGMIX (Hendrycks et al., 2020), REGMIX (Pinto et al., 2023), VOS (Du et al., 2022) and our proposed BER. OpenCIL is based on four CIL models, including two regularization-based methods – iCaRL (Rebuffi et al., 2017) and WA (Zhao et al., 2020), one replay-based method BiC (Wu et al., 2019), and one parameter-isolation-based method FOSTER (Wang et al., 2022a). More details are presented in Appendix B.

To ensure a fair comparison across methods originating from different areas, we use unified settings with common hyperparameters and architecture choices. Following the most commonly used architecture for the respective ID dataset in the CIL community (Wang et al., 2022a; Zhao et al., 2020), the backbone ResNet32 is used when CIFAR100 is used as the ID dataset, and ResNet18 is used whenever ImageNet1K is the ID dataset. All results are averaged over three independent runs using different random seeds. More details are presented in Appendix C.

### 3.2 MAIN FINDINGS

We summarize our main findings and justification based on our OpenCIL benchmarking as follows.

*OOD detectors suffer from catastrophic forgetting as well when applied to CIL models.* This is illustrated in Fig. 3a, where the performance of different OOD detection methods decreases significantly with increasing incremental steps. This is because, with more incremental steps, more old class samples are wrongly detected as OOD samples, while at the same time, more OOD samples are misclassified into new classes, leading to fast downgraded OOD detection performance.

*However, catastrophic forgetting is more persistent in CIL models than OOD detection methods.* As shown in Fig. 3b, the AUC performance for OOD detection drops fast in early incremental steps, and then it slows down and levers off, but the ACC performance for CIL has a continuous, fast decrease throughout all incremental steps. This is because remembering the forgotten ID samples is more difficult than distinguishing them from OOD data. Also, after the CIL models reach a certain level of forgetting for some ID classes, the performance of distinguishing them from OOD data tends to be stable, resulting in relatively stable OOD detection performance at the end of incremental learning.

*Consequently, CIL models are prone to misclassifying OOD samples into new classes.* This is also shown in Fig. 1b, where the CIL models often yield significantly higher prediction confidence on misclassifying the OOD samples into the new classes than the old classes, due to the presence of large samples from these new classes at each incremental step. The reason is that due to more severe catastrophic forgetting in CIL, the CIL models predict the ID samples as the new classes with higher confidence than that for the old classes and the OOD samples.

*In addition to catastrophic forgetting, CIL models need to handle new issues in the presence of OOD samples, since they exhibit stronger prediction confidence on OOD samples than Old class samples.* As shown in Figs. 1a and 1b, the ID samples from old classes/tasks often have lower prediction confidence than different OOD samples, thereby being misclassified as the OOD samples. This phenomenon becomes more severe with an increasing number of incremental steps. This is because the CIL models tend to be less confident when predicting the samples of old classes due to the CF problem, making their OOD scores lower than samples from the OOD data.

*On the other hand, CIL models with fine-tuning-based OOD methods show to be more advantageous than those with post-hoc-based methods.* This can be observed by looking at the average AUC and FPR results of each CIL model for the nine post-hoc-based and the five previous fine-tuning methods in Tables 1, 2, and 3. The observation holds for both ID datasets. This demonstrates the advantage of tuning an additional classifier for OOD detection, but note that it is at the expense of some computational overhead. Besides, CIL models with higher CIL accuracy often gain better OOD detection performance. This can be observed in Tables 1, 2, and 3, where, when averaged over all the existing OOD methods used, the CIL models with higher CIL accuracy, *e.g.*, WA and FOSTER, often achieve better AUC and FPR performance than the other two CIL models, especially on CIFAR100. This observation is consistent with the performance at each incremental step, as shown in Fig. 3c. This is because better CIL algorithms can keep more essential information about ID data to improve ID classification, which can also prevent OOD data from being misclassified into ID classes.

## 4 OUR PROPOSED APPROACH BER

As summarized above, one key issue for the CIL models in the presence of OOD samples is the increasing biases of the CIL models towards OOD samples and newly added classes with the growth of incremental steps due to more severe catastrophic forgetting. Further, fine-tuning-based OOD methods are generally more effective than the post-hoc methods. Therefore, we introduce the novel approach, **Bi-directional Energy Regularization (BER)**, a fine-tuning-based OOD detection approach, to tackle this bias issue, *i.e.*, avoiding the classification of the old and new class samples as OOD samples. BER consists of two components, namely *New Task Energy Regularization (NTER)* and *Old Task Energy Regularization (OTER)*. NTER is designed to distinguish OOD data from samples of new task classes, while OTER is designed to distinguish OOD data from samples of old task classes. Below we introduce each component in detail.

### 4.1 NEW TASK ENERGY REGULARIZATION (NTER)

Due to the overwhelming presence of new class samples, CIL models typically demonstrate a strongly biased prediction on the OOD samples towards the new classes, *i.e.*, they have high prediction confidence on classifying OOD samples into the classes in the new task, as illustrated in Fig. 1b. To address this issue, NTER synthesizes the pseudo OOD samples that are distributed on the decision boundary of different new classes, and further utilizes them to enlarge the decision boundary margin, as illustrated in Figure 2 **Right**. Specifically, NTER first randomly mixes up samples of different new classes as the pseudo-OOD samples. Formally, let $\mathbf{x}_t = \{x_t^1, x_t^2, ..., x_t^b, \}, \mathbf{x}_t \in X_t^{train}$ be one training batch of ID data from new classes at task $t$, where task $t$ is the new task and $b$ is the batch size, $\mathbf{y}_t = \{y_t^1, y_t^2, ..., y_t^b, \}$ and $\mathbf{y}_t \in Y_t$ be the corresponding class label of $\mathbf{x}_t$, then a pseudo-OOD sample $\bar{x}_t$ is synthesized as follows:

$$\bar{x}_t = \beta x_t^i + (1 - \beta)x_t^j, y_t^i \neq y_t^j, \tag{2}$$

where $\beta \in [0, 1]$ is sampled from Beta distribution. Motivated by the success of energy-based methods (Liu et al., 2020), we further utilize these pseudo-OOD samples to regularize the classification

of new class samples via the following energy loss function:

$$\mathcal{L}_n = \mathbb{E}_{x_t \sim X_t^{train}}[(max(0, p_{in} - E(x_t)))^2] + \mathbb{E}_{(x_t^i, x_t^j) \sim X_t^{train}}[(max(0, E(\bar{x}_t) - p_{out}))^2], \quad (3)$$

where $E(x; f) = -\tau \cdot log(\sum_{j=1}^{Q_t} e^{f_j(x)/\tau})$, $Q_t$ is the whole label space that the class set of all seen classes at task $t$, and $\tau$ is a temperature scaling hyperparameter. Note that we do not combine all possible pairs of $(x_t^i, x_t^j)$ in the full training dataset $X_t^{train}$ to form $\bar{x}_t$, which are produced within the mini-batches for efficiency consideration (Zhou et al., 2021). As a result, the time complexity is of the same magnitude as vanilla training, having minimal computational overhead.

### 4.2 Old Task Energy Regularization (OTER)

Due to the severe catastrophic forgetting problem, as illustrated in Fig. 1a, the CIL models exhibit significantly lower prediction confidence on samples of old classes than new classes. As a result, the CIL models often misclassify the old class samples as the OOD samples. To address this issue, OTER performs a different mixup operation from NTER to generate more samples of old classes. In particular, it randomly mixes up new class samples with old class samples to synthesize old class samples. This mixup operation not only increases the diversity of the small-sized old class samples but also transfers information from new class samples to old class samples. Then using these mixup samples in the fine-tuning stage largely enhances the prediction confidence of the old class samples and expands their decision boundary, as illustrated in Figure 2 **Right**. Formally, let $\mathbf{x}_t = \{x_t^1, x_t^2, ..., x_t^b,\}, \mathbf{x}_t \in X_t^{train}$ be one training batch of ID data from new classes at task $t$, where task $t$ is the new task and $b$ is the batch size, $M_t = \{m_t^1, m_t^2, ..., m_t^s\}$ be the samples of old classes in the memory bank for task $t$, and $s$ is the memory size, then the synthesize old class samples $\bar{m}_t$ are generated via:

$$\bar{m}_t = \lambda x_t + (1 - \lambda)m_t, \quad (4)$$

where $\lambda$ is a mixup hyperparameter, similar to $\beta$ in Eq. 2. We further leverage these augmented ID samples to boost the prediction confidence of old class samples via the following energy loss function:

$$\mathcal{L}_o = \mathbb{E}_{(x_t, m_t) \sim (X_t^{train}, M_t)}[(max(0, E(\bar{m}_t) - p_{in}))^2], \quad (5)$$

where $E(x; f) = -\tau \cdot log(\sum_{j=1}^{Q_t} e^{f_j(x)/\tau})$, $Q_t$ is the whole label space that the class set of all seen classes at task $t$, and $\tau$ is the temperature scaling. Note that we only apply energy regularization to the ID data in OTER, without the energy regularization term on the pseudo-OOD samples in Eq. 3. The old class samples cannot be used for synthesizing the pseudo-OOD samples via, *e.g.*, mixup between old and new class samples, or mixup between samples of different old classes. This is because, as the samples stored in the data replay memory, these old task samples are the most representative samples of the old classes, which are typically located near the class center in their respective belonging classes in the feature space. Therefore, generating pseudo-OOD samples using these old class samples will severely compress the decision boundary of old task classes, resulting in a significant adverse impact on the OOD detection performance.

Lastly, we utilize the cross-entropy loss, together with the two energy regularization losses, to fine-tune the extra classifier $f_t(\cdot)$ for the CIL models following the fine-tuning OOD detection framework. Thus, the overall optimization objective of our BER approach at each task $t$ is as follows:

$$\mathcal{L} = \mathbb{E}_{(x,y) \sim (T_t^{train}, Q_t)}[\ell(f(x), y] + \alpha(\mathcal{L}_n + \mathcal{L}_o), \quad (6)$$

where $\ell$ is a cross-entropy loss, $\alpha$ is the hyperparameter, $\mathcal{L}_n$ is as defined in Eq. 3, and $\mathcal{L}_o$ is as defined in Eq. 5. During inference, BER utilizes the Energy Score defined in (Liu et al., 2020) as the OOD score. The overall algorithm of our proposed BER is provided in Appendix F.

### 4.3 Empirical Evaluation

We perform large-scale experiments that evaluate the composition of the proposed BER-based OOD detection method and four CIL models using three different incremental step sizes based on two ID datasets CIFAR100 and ImageNet1K. To ensure a fair comparison, BER is incorporated into the four CIL models in exactly the same way as the other fine-tuning-based methods. Due to the space limitation, we only report the average OOD detection metric values (AUC, FPR) on all six OOD datasets for each ID dataset, more fine-grained results (on near-OOD datasets and far-OOD datasets, respectively) and more OOD detection metric AP values are presented in Appendix E.

Table 1: Main results on OpenCIL benchmark for the **step size of** $k = 5$ **for CIFAR 100** and the **step size of** $k = 50$ **for ImageNet1K**. The results are average over six OOD datasets and all incremental steps. Either the post-hoc-based or fine-tuning-based OOD methods do not affect the original CIL performance, so their average CIL accuracy on the ID data remains the same. The **best** and second-best performance per dataset in the fine-tuning-based methods are highlighted.

| | ID Dataset: CIFAR100 | | | | | | | | | | ID Dataset: ImageNet1K | | | | | | | | | |
| | iCaRL | | BiC | | WA | | FOSTER | | Average | | iCaRL | | BiC | | WA | | FOSTER | | Average | |
| | AUC↑ | FPR↓ | AUC↑ | FPR↓ | AUC↑ | FPR↓ | AUC↑ | FPR↓ | AUC↑ | FPR↓ | AUC↑ | FPR↓ | AUC↑ | FPR↓ | AUC↑ | FPR↓ | AUC↑ | FPR↓ | AUC↑ | FPR↓ |
|---|---|---|---|---|---|---|---|---|---|---|---|---|---|---|---|---|---|---|---|---|
| *Average CIL accuracy* | 58.20 | | 55.87 | | 61.44 | | 63.51 | | 59.76 | | 40.86 | | 42.26 | | 45.99 | | 45.96 | | 43.77 | |
| *Post-hoc-based OOD methods* | | | | | | | | | | | | | | | | | | | | |
| MSP | 66.89 | 87.86 | 68.66 | 87.22 | 69.19 | 86.11 | 70.01 | 85.68 | 68.69 | 86.72 | 61.87 | 91.03 | 63.06 | 92.46 | 61.69 | 92.85 | 65.45 | 90.89 | 63.02 | 91.81 |
| ODIN | 70.26 | 79.60 | 70.78 | 78.54 | 71.70 | 80.09 | 72.89 | 75.48 | 71.41 | 78.43 | 66.00 | 88.90 | 67.62 | 90.54 | 63.47 | 91.07 | 69.16 | 87.58 | 66.56 | 89.52 |
| Energy | 70.23 | 81.17 | 69.55 | 82.78 | 71.99 | 82.53 | 73.89 | 76.34 | 71.44 | 80.70 | 62.97 | 91.95 | 65.61 | 92.43 | 62.43 | 92.94 | 67.46 | 90.94 | 64.62 | 92.06 |
| MaxLogit | 70.16 | 81.87 | 69.87 | 83.31 | 71.89 | 82.96 | 73.82 | 77.34 | 71.44 | 81.37 | 63.94 | 91.58 | 64.67 | 93.32 | 63.51 | 91.44 | 67.31 | 91.30 | 64.86 | 91.91 |
| GEN | 70.39 | 82.07 | 70.89 | 79.54 | 72.22 | 82.14 | 74.21 | 76.95 | 71.93 | 80.17 | 55.67 | 93.51 | 60.65 | 92.82 | 61.27 | 93.38 | 59.94 | 94.26 | 59.38 | 93.49 |
| ReAct | 70.21 | 81.26 | 69.86 | 84.89 | 73.54 | 82.15 | 74.29 | 76.97 | 71.97 | 81.32 | 55.67 | 93.51 | 60.64 | 92.82 | 61.27 | 93.37 | 59.94 | 94.26 | 59.38 | 93.46 |
| KLM | 66.21 | 88.89 | 67.53 | 86.71 | 68.19 | 88.70 | 69.38 | 86.36 | 67.83 | 87.42 | 63.34 | 89.99 | 62.93 | 89.79 | 63.18 | 89.52 | 66.33 | 87.95 | 63.95 | 89.31 |
| Relation | 66.33 | 78.06 | 70.64 | 81.42 | 71.89 | 77.19 | 72.49 | 75.74 | 70.34 | 78.10 | 63.13 | 89.86 | 66.39 | 93.63 | 63.11 | 95.41 | 63.48 | 90.83 | 64.03 | 92.43 |
| NNGuide | 70.27 | 78.83 | 70.70 | 79.64 | 71.60 | 79.00 | 73.68 | 75.96 | 71.56 | 78.36 | 63.00 | 89.47 | 69.87 | 85.81 | 62.61 | 90.30 | 68.63 | 88.54 | 66.03 | 88.53 |
| Average | 68.99 | 82.18 | 69.83 | 82.67 | 71.36 | 82.21 | 72.74 | 78.54 | 70.73 | 81.40 | 62.58 | 90.70 | 64.86 | 91.18 | 62.72 | 91.83 | 66.01 | 90.03 | 64.04 | 90.94 |
| *Fine-tuning-based OOD methods* | | | | | | | | | | | | | | | | | | | | |
| LogitNorm | 70.21 | 81.18 | 69.22 | 83.39 | 71.13 | 82.65 | 73.31 | 76.78 | 70.97 | 81.00 | 62.02 | 92.73 | 66.30 | 91.20 | 61.58 | 93.94 | 65.87 | 93.26 | 63.94 | 92.78 |
| T2FNorm | 70.45 | 81.50 | 69.59 | 83.29 | 70.90 | 83.26 | 73.26 | 77.35 | 71.05 | 81.35 | 62.82 | 92.12 | 65.80 | 92.30 | 62.40 | 92.84 | 66.55 | 92.01 | 64.39 | 92.32 |
| AUGMIX | 70.27 | 81.22 | 68.65 | 83.16 | 71.11 | 82.93 | 73.12 | 76.98 | 70.79 | 81.07 | 62.19 | 90.86 | 65.97 | 89.82 | 62.88 | 90.12 | 67.53 | 89.45 | 64.64 | 90.06 |
| REGMIX | 70.92 | 81.33 | 68.99 | 83.65 | 71.44 | 84.30 | 73.78 | 77.56 | 71.28 | 81.71 | 63.93 | 91.20 | 66.24 | 90.29 | 62.96 | 91.56 | 67.84 | 91.56 | 65.24 | 91.17 |
| VOS | 71.54 | 79.83 | 67.63 | 77.80 | 66.73 | 81.28 | 72.70 | 77.08 | 69.65 | 79.00 | 61.33 | 89.74 | 65.74 | 90.22 | 62.08 | 89.98 | 66.73 | 90.14 | 63.97 | 90.02 |
| **BER (Ours)** | **72.75** | **77.59** | **71.47** | **77.82** | **72.47** | **78.69** | **74.20** | **74.93** | **72.72** | **77.26** | **63.45** | **89.53** | **67.72** | **88.61** | **64.09** | **89.15** | **69.34** | **88.39** | **66.15** | **88.92** |
| Average | 70.68 | 81.01 | 68.82 | 82.26 | 70.26 | 82.88 | 73.23 | 77.15 | 70.75 | 80.83 | 62.46 | 91.33 | 66.01 | 90.77 | 62.38 | 91.70 | 66.90 | 91.28 | 64.44 | 91.27 |
| Average (All) | 69.59 | 81.76 | 69.47 | 82.52 | 70.97 | 82.45 | 72.91 | 78.04 | 70.74 | 81.20 | 62.54 | 90.92 | 65.27 | 91.03 | 62.60 | 91.78 | 66.33 | 90.48 | 64.18 | 91.06 |

Table 2: Main results on OpenCIL benchmark for the **step size of** $k = 10$ **for CIFAR 100** and the **step size of** $k = 100$ **for ImageNet1K**. The results are averaged over six OOD datasets and all incremental steps. Either the post-hoc-based or fine-tuning-based OOD methods do not affect the original CIL performance, so their average CIL accuracy on the ID data remains the same. The **best** and second-best performance per dataset in the fine-tuning-based methods are highlighted.

| | ID Dataset: CIFAR100 | | | | | | | | | | ID Dataset: ImageNet1K | | | | | | | | | |
| | iCaRL | | BiC | | WA | | FOSTER | | Average | | iCaRL | | BiC | | WA | | FOSTER | | Average | |
| | AUC↑ | FPR↓ | AUC↑ | FPR↓ | AUC↑ | FPR↓ | AUC↑ | FPR↓ | AUC↑ | FPR↓ | AUC↑ | FPR↓ | AUC↑ | FPR↓ | AUC↑ | FPR↓ | AUC↑ | FPR↓ | AUC↑ | FPR↓ |
|---|---|---|---|---|---|---|---|---|---|---|---|---|---|---|---|---|---|---|---|---|
| *Average CIL accuracy* | 60.08 | | 61.68 | | 65.88 | | 66.01 | | 63.41 | | 44.44 | | 49.63 | | 52.22 | | 52.29 | | 49.65 | |
| *Post-hoc-based OOD methods* | | | | | | | | | | | | | | | | | | | | |
| MSP | 67.77 | 88.33 | 66.67 | 86.98 | 70.49 | 86.04 | 71.69 | 84.94 | 69.16 | 86.57 | 63.84 | 89.81 | 67.45 | 91.80 | 65.97 | 89.17 | 67.64 | 89.32 | 66.23 | 90.03 |
| ODIN | 70.05 | 81.93 | 69.76 | 80.52 | 71.86 | 80.18 | 74.70 | 75.68 | 71.59 | 79.58 | 68.41 | 87.91 | 71.09 | 87.45 | 67.86 | 89.08 | 70.24 | 88.15 | 69.40 | 88.15 |
| Energy | 70.35 | 83.74 | 69.76 | 82.01 | 73.03 | 81.22 | 75.76 | 77.24 | 72.23 | 81.05 | 65.64 | 91.88 | 70.16 | 89.57 | 66.57 | 92.22 | 69.39 | 90.49 | 67.94 | 91.04 |
| MaxLogit | 70.34 | 84.45 | 69.54 | 82.78 | 72.98 | 82.33 | 75.71 | 78.11 | 72.14 | 81.92 | 65.83 | 91.38 | 69.93 | 89.42 | 66.75 | 92.03 | 69.56 | 90.02 | 68.02 | 90.71 |
| GEN | 70.86 | 84.31 | 69.87 | 82.51 | 73.30 | 81.42 | 76.11 | 77.64 | 72.54 | 81.47 | 66.89 | 90.78 | 69.51 | 89.77 | 66.66 | 90.94 | 69.74 | 89.91 | 68.67 | 89.75 |
| ReAct | 68.79 | 84.43 | 70.62 | 82.60 | 74.87 | 81.82 | 76.15 | 78.03 | 72.61 | 81.72 | 55.66 | 94.47 | 66.57 | 89.86 | 63.85 | 90.94 | 65.15 | 91.16 | 62.81 | 91.61 |
| KLM | 67.51 | 88.00 | 66.27 | 87.66 | 69.89 | 87.65 | 71.34 | 85.90 | 68.75 | 87.30 | 66.48 | 88.28 | 67.32 | 87.17 | 66.39 | 88.50 | 70.23 | 88.45 | 67.32 | 88.37 |
| Relation | 64.40 | 89.57 | 65.48 | 80.85 | 76.12 | 88.37 | 72.17 | 75.29 | 69.54 | 83.52 | 66.38 | 89.92 | 69.10 | 89.63 | 67.51 | 94.59 | 67.93 | 88.99 | 67.73 | 90.78 |
| NNGuide | 72.14 | 79.65 | 68.74 | 78.79 | 75.60 | 76.00 | 75.90 | 74.70 | 73.09 | 77.28 | 66.93 | 89.88 | 70.64 | 87.33 | 68.69 | 88.82 | 67.63 | 88.32 | 68.37 | 88.59 |
| Average | 69.13 | 84.93 | 68.52 | 82.74 | 73.13 | 82.78 | 74.39 | 78.61 | 71.29 | 82.27 | 65.07 | 90.48 | 69.09 | 89.11 | 66.78 | 90.55 | 68.61 | 89.42 | 67.39 | 89.89 |
| *Fine-tuning-based OOD methods* | | | | | | | | | | | | | | | | | | | | |
| LogitNorm | 70.44 | 83.78 | 70.64 | 82.46 | 72.04 | 81.33 | 74.74 | 78.28 | 71.97 | 81.46 | 65.15 | 92.51 | 69.54 | 87.26 | 66.30 | 93.16 | 67.84 | 92.39 | 67.21 | 91.33 |
| T2FNorm | 70.65 | 84.14 | 71.75 | 82.72 | 71.98 | 82.33 | 74.65 | 78.83 | 72.26 | 82.00 | 66.09 | 91.58 | 69.53 | 89.09 | 67.39 | 91.83 | 68.19 | 90.95 | 67.80 | 90.86 |
| AUGMIX | 70.72 | 83.91 | 70.78 | 82.94 | 72.11 | 81.53 | 74.75 | 78.62 | 72.09 | 81.75 | 66.48 | 90.69 | 70.83 | 87.98 | 68.10 | 89.73 | 69.62 | 88.53 | 69.29 | 88.43 |
| REGMIX | 71.47 | 84.53 | 71.43 | 82.23 | 72.01 | 82.95 | 75.26 | 79.35 | 72.54 | 82.26 | 66.41 | 90.69 | 69.83 | 89.68 | 67.45 | 90.69 | 69.43 | 90.78 | 68.28 | 90.46 |
| VOS | 72.31 | 82.53 | 66.66 | 79.03 | 68.49 | 78.15 | 74.58 | 77.71 | 70.51 | 79.36 | 66.28 | 91.67 | 66.67 | 89.47 | 65.41 | 89.92 | 68.98 | 88.74 | 66.83 | 89.95 |
| **BER (Ours)** | **74.17** | **78.87** | **71.83** | **77.76** | **76.77** | **75.58** | **76.40** | **74.58** | **74.79** | **76.70** | **68.80** | **87.51** | **71.94** | **86.80** | **69.06** | **87.59** | **70.54** | **87.44** | **70.09** | **87.33** |
| Average | 71.12 | 83.78 | 70.25 | 81.88 | 71.33 | 81.26 | 74.80 | 78.56 | 71.87 | 81.37 | 66.51 | 90.89 | 69.28 | 88.67 | 66.93 | 90.99 | 68.81 | 90.28 | 67.88 | 90.21 |
| Average (All) | 69.84 | 84.52 | 69.14 | 82.43 | 72.49 | 82.24 | 74.54 | 78.59 | 71.50 | 81.95 | 65.58 | 90.63 | 69.16 | 88.95 | 66.83 | 90.71 | 68.68 | 89.73 | 67.56 | 90.00 |

**Performance of BER.** We compare the OOD detection capability of our proposed BER with the five fine-tuning-based OOD detectors among four CIL models based on two ID datasets CIFAR100 and ImageNet1K. Tables 1, 2, and 3 present the comparisons at the step size of $k = \{5, 10, 20\}$ for CIFAR100 on left side and at the step size of $k = \{50, 100, 200\}$ for ImageNet1K on right side, respectively. For the two metrics of four CIL models on CIFAR100 across three steps, there are 24 possible combinations for each OOD detector. In 23 out of 24 cases, BER achieves the best OOD detection performance across the combination of four representative CIL models and three types of step sizes in AUR and FPR. As for ImageNet1K, BER also achieves the best OOD detection performance in 21 out of 24 cases. Note that the other three best performers on ImageNet1K are achieved by three different previous OOD detectors combined with different CIL models. BER also achieves the second-best performance in the cases where it is not the best performer. This demonstrates the strong robustness of our proposed BER under diverse combined CIL and OOD dataset and model scenarios. Furthermore, BER also achieves the best performance in the average results across four CIL models for all three types of step sizes in AUC and FPR on both the CIFAR100 and ImageNet1K datasets. This consistent improvement indicates that our energy regularization on both the old and new classes helps effectively mitigate biases towards the OOD samples and new classes in the use of fine-tuning-based OOD detection methods in CIL models.

Table 3: Main results on OpenCIL benchmark for the **step size of** $k = 20$ **for CIFAR 100** and the **step size of** $k = 200$ **for ImageNet1K**. The results are averaged over six OOD datasets and all incremental steps. Either the post-hoc-based or fine-tuning-based OOD methods do not affect the original CIL performance, so their average CIL accuracy on the ID data remains the same. The **best** and second-best performance per dataset in the fine-tuning-based methods are highlighted.

| | ID Dataset: CIFAR100 | | | | | | | | | | ID Dataset: ImageNet1K | | | | | | | | | |
| | iCaRL | | BiC | | WA | | FOSTER | | Average | | iCaRL | | BiC | | WA | | FOSTER | | Average | |
| | AUC↑ | FPR↓ | AUC↑ | FPR↓ | AUC↑ | FPR↓ | AUC↑ | FPR↓ | AUC↑ | FPR↓ | AUC↑ | FPR↓ | AUC↑ | FPR↓ | AUC↑ | FPR↓ | AUC↑ | FPR↓ | AUC↑ | FPR↓ |
|---|---|---|---|---|---|---|---|---|---|---|---|---|---|---|---|---|---|---|---|---|
| *Average CIL accuracy* | | | | | | | | | | | | | | | | | | | | |
| | 62.65 | | 64.14 | | 68.05 | | 68.75 | | 65.90 | | 48.85 | | 53.30 | | 58.42 | | 57.84 | | 54.60 | |
| *Post-hoc-based OOD methods* | | | | | | | | | | | | | | | | | | | | |
| MSP | 68.38 | 87.19 | 69.72 | 87.75 | 72.46 | 84.74 | 72.10 | 85.02 | 70.66 | 86.17 | 66.83 | 87.47 | 71.39 | 86.68 | 70.73 | 87.12 | 71.57 | 84.44 | 70.13 | 86.43 |
| ODIN | 71.34 | 81.20 | 71.46 | 80.62 | 74.27 | 78.08 | 74.45 | 76.73 | 72.88 | 79.16 | 71.36 | 83.61 | 75.27 | 82.55 | 71.69 | 84.60 | 73.56 | 82.93 | 72.97 | 83.42 |
| Energy | 72.38 | 81.99 | 71.48 | 82.45 | 75.22 | 79.00 | 76.14 | 78.54 | 73.81 | 80.50 | 69.52 | 88.30 | 74.36 | 84.92 | 70.56 | 87.73 | 72.67 | 87.09 | 71.78 | 87.01 |
| MaxLogit | 72.32 | 82.20 | 71.60 | 83.73 | 75.22 | 80.02 | 76.08 | 78.84 | 73.80 | 81.20 | 69.62 | 87.90 | 74.17 | 85.99 | 71.08 | 87.26 | 73.15 | 85.57 | 72.00 | 86.68 |
| GEN | 72.73 | 82.44 | 71.92 | 83.00 | 75.36 | 79.38 | 76.55 | 78.39 | 74.14 | 80.80 | 70.16 | 87.60 | 74.19 | 85.84 | 71.68 | 85.67 | 73.84 | 84.30 | 72.47 | 85.85 |
| ReAct | 71.59 | 82.99 | 73.69 | 82.87 | 76.82 | 78.49 | 77.33 | 78.85 | 74.86 | 80.80 | 59.48 | 92.57 | 69.67 | 85.93 | 63.12 | 88.28 | 64.82 | 89.13 | 64.27 | 88.98 |
| KLM | 69.10 | 85.53 | 70.19 | 86.76 | 72.47 | 86.06 | 72.47 | 84.39 | 71.06 | 85.69 | 69.51 | 84.99 | 70.55 | 85.72 | 69.61 | 87.49 | 73.55 | 82.10 | 70.81 | 85.07 |
| Relation | 65.37 | 80.08 | 69.10 | 79.46 | 76.23 | 77.46 | 71.27 | 76.53 | 70.49 | 78.38 | 71.24 | 87.09 | 75.12 | 83.34 | 71.45 | 92.75 | 73.27 | 83.31 | 72.77 | 86.62 |
| NNGuide | 74.14 | 77.05 | 71.24 | 78.89 | 76.28 | 73.78 | 76.20 | 76.13 | 74.47 | 76.46 | 71.41 | 88.03 | 77.95 | 86.53 | 71.90 | 86.90 | 73.27 | 82.74 | 73.63 | 86.05 |
| Average | 70.82 | 82.30 | 71.16 | 82.84 | 74.93 | 79.67 | 74.73 | 79.27 | 72.91 | 81.02 | 68.79 | 87.51 | 73.63 | 85.28 | 70.20 | 87.53 | 72.19 | 84.62 | 71.20 | 86.24 |
| *Fine-tuning-based OOD methods* | | | | | | | | | | | | | | | | | | | | |
| LogitNorm | 72.41 | 82.32 | 71.27 | 83.80 | 74.29 | 79.49 | 75.38 | 79.28 | 73.34 | 81.22 | 69.40 | 89.17 | 73.18 | 84.05 | 69.14 | 89.57 | 71.62 | 89.30 | 70.84 | 88.02 |
| T2FNorm | 72.67 | 82.39 | 71.51 | 83.79 | 74.38 | 80.66 | 75.70 | 79.60 | 73.56 | 81.61 | 70.45 | 87.47 | 74.21 | 82.17 | 70.95 | 87.36 | 71.64 | 87.72 | 71.81 | 86.18 |
| AUGMIX | 72.50 | 82.62 | 70.70 | 84.33 | 74.35 | 79.97 | 75.54 | 79.61 | 73.27 | 81.63 | 71.32 | 87.91 | 76.72 | 84.63 | 70.84 | 87.20 | 72.81 | 83.66 | 72.92 | 85.85 |
| REGMIX | 72.74 | 82.81 | 71.62 | 84.65 | 74.78 | 80.45 | 75.69 | 80.09 | 73.71 | 82.00 | 69.56 | 89.61 | 72.78 | 87.57 | 70.68 | 88.13 | 71.84 | 88.85 | 71.22 | 88.54 |
| VOS | 72.93 | 81.39 | 68.77 | 78.28 | 72.21 | 76.57 | 75.07 | 78.52 | 72.24 | 78.69 | 62.94 | 89.63 | 64.19 | 88.20 | 62.86 | 89.02 | 65.61 | 88.09 | 63.90 | 88.73 |
| **BER (Ours)** | 74.55 | 76.75 | 71.90 | 77.97 | 76.64 | 71.72 | 76.52 | 75.04 | 74.90 | 75.37 | 71.88 | 87.00 | 76.46 | 83.01 | 71.98 | 86.24 | 73.95 | 82.66 | 73.57 | 84.73 |
| Average | 72.65 | 82.31 | 70.77 | 82.97 | 74.00 | 79.43 | 75.48 | 79.42 | 73.22 | 81.03 | 68.73 | 88.76 | 72.22 | 85.32 | 68.89 | 88.26 | 70.70 | 87.52 | 70.14 | 87.46 |
| Average (All) | 71.47 | 82.30 | 71.02 | 82.89 | 74.60 | 79.58 | 75.00 | 79.32 | 73.02 | 81.02 | 68.77 | 87.96 | 73.13 | 85.29 | 69.73 | 87.79 | 71.66 | 85.66 | 70.82 | 86.68 |

Table 4: Ablation study of our proposed BER based on the CIFAR100 ID dataset with the step size of $k = 10$. Energy (Liu et al., 2020) is used as the baseline that does not use both NTER and OTER.

| Loss Component | | Near OOD Datasets | | | | Far OOD Datasets | | | | | | | |
| NTER | OTER | CIFAR10 | | TIN | | MNIST | | SVHN | | Texture | | Places365 | |
| | | AUC↑ | FPR↓ | AUC↑ | FPR↓ | AUC↑ | FPR↓ | AUC↑ | FPR↓ | AUC↑ | FPR↓ | AUC↑ | FPR↓ |
|---|---|---|---|---|---|---|---|---|---|---|---|---|---|
| ✗ | ✗ | 70.70 | 84.86 | 72.74 | 83.12 | 86.63 | 62.37 | 62.56 | 96.81 | 56.18 | 92.92 | 73.31 | 82.33 |
| ✓ | ✗ | 71.31 | 84.16 | 73.64 | 82.37 | 89.93 | 52.73 | 65.63 | 95.93 | 59.66 | 91.15 | 75.33 | 79.72 |
| ✗ | ✓ | 70.77 | 83.84 | 73.80 | 82.10 | 88.65 | 55.36 | 64.67 | 95.80 | 59.26 | 91.94 | 74.35 | 81.34 |
| ✓ | ✓ | **71.60** | **83.17** | **75.84** | **81.06** | **91.56** | **46.02** | **67.52** | **93.08** | **62.59** | **90.52** | **75.90** | **79.35** |

**Ablation Study.** This section evaluates the importance of the two key components of BER, New Task Energy Regularization (NTER) and Old Task Energy Regularization (OTER). Table 4 presents the results of the ablation study conducted on these two components using six OOD datasets individually based on the CIFAR100 ID dataset with the step size of $k = 10$ using Energy (Liu et al., 2020) as the baseline. The results show that either NTER or OTER helps boost the AUC performance and reduce the FPR on both near and far OOD datasets, and they can achieve the best performance when the two components are combined. Since NTER is designed to alleviate the misclassification of OOD samples into new classes while OTER is designed to reduce the misclassification of old class samples as OOD samples, their combination results in a detection model that largely reduces both types of detection errors.

## 5 CONCLUSION

In this paper, we introduce OpenCIL, the first large-scale and systematic benchmark designed to enable CIL models with existing OOD detectors, regarding the CIL models in open world applications. OpenCIL introduces two principled frameworks for incorporating diverse OOD detectors into CIL models and a new evaluation pipeline for fairly evaluating the capability of OOD detectors in incremental learning. In particular, OpenCIL accommodates four representative CIL models with 15 diverse OOD detection methods, resulting in 60 baseline models on two popular CIL datasets and six commonly-used near/far OOD datasets. Based on our large-scale experiments on OpenCIL, we offer several important insights into the design of CIL models for open-world applications. We further propose a novel approach, namely Bi-directional Energy Regularization (BER), which utilizes energy regularization based on two types of sample synthesis to effectively mitigate the increasing bias of CIL Models towards OOD samples and newly added classes with the growth of incremental steps. Extensive experiments demonstrate that BER achieves state-of-the-art performance on the OpenCIL benchmark under varying incremental step sizes on popular CIL and OOD datasets, improving the OOD detection capability of a wide range of CIL models.

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

## A    MORE EVALUATION SETTING DETAILS

### A.1    DATASETS

The in-distribution (ID) datasets for class incremental learning (CIL) include two datasets. 1) CI-FAR100 (Krizhevsky et al., 2009) contains $50,000$ training images and $10,000$ test images of size $32 \times 32$ with 100 classes, and 2) ImageNet (Russakovsky et al., 2015) is a large-scale classification dataset, which contains 1.28M training images and 50K testing images with $1,000$ classes sampled from nature images.

We select six commonly used out-of-distribution (OOD) datasets for OOD detection on CIFRA100. Following the recent large-scale OOD detection benchmark OpenOOD (Yang et al., 2022), our OOD datasets include two near OOD datasets: CIFAR10 (Krizhevsky et al., 2009) and Tiny-ImageNet (TIN) (Le & Yang, 2015), and four far OOD datasets: MNIST (LeCun et al., 2010), Texture (Cimpoi et al., 2014), Places365 (Zhou et al., 2017), and SVHN (Netzer et al., 2011). CIFAR10 (Krizhevsky et al., 2009) contains $10,000$ images with 10 classes. TIN (Le & Yang, 2015) contains $7,498$ images, which removes the $2,502$ images that have overlapping semantics (Yang et al., 2021) with CIFAR100 classes. MNIST (LeCun et al., 2010) contains $10,000$ images with 10 classes. Texture (Cimpoi et al., 2014) contains $5,640$ images with 47 classes. Places365 (Zhou et al., 2017) contains $33,773$ images, which removes the $2,727$ images that have overlapping semantics (Yang et al., 2021) with CIFAR100 classes. SVHN (Netzer et al., 2011) contains $26,032$ images with 10 class.

We also select six commonly used OOD datasets for OOD detection on ImageNet1k. Following OpenOOD (Yang et al., 2022), these includes four near OOD datasets: ImageNet_O (Hendrycks et al., 2021), iNaturalist (Van Horn et al., 2018), OpenImage_O (Wang et al., 2022b) and Species (Basart et al., 2022), and two far OOD datasets: MNIST (LeCun et al., 2010) and Texture (Cimpoi et al., 2014). ImageNet_O (Hendrycks et al., 2021) contains $2,000$ images from categories not found in the ImageNet1k dataset. We use a $10,000$ image subset of iNaturalist (Van Horn et al., 2018), which is based on 110 manually selected plant classes not present in ImageNet1k. The OOD samples are randomly sampled images from these 110 classes (Huang & Li, 2021). All images are resized to have a max dimension of 800 pixels. OpenImage_O (Wang et al., 2022b) contains $15,869$ images with the support of a manual filter. We use a $10,000$ subset of 713K images Species (Basart et al., 2022) with 10 classes. Two far OOD datasets are the same as CIFAR100. Near OOD datasets mean that their OOD samples have small semantic shifts compared with the ID samples, while far OOD datasets mean that their OOD samples are very different from the ID samples, typically containing obvious covariate (domain) shift (Yang et al., 2022). A summary of the CIL ID and OOD datasets is presented in Table 5.

Table 5: Key statistics of used CIL ID datasets and OOD datasets.

| Benchmark | CIFAR100 | | | ImageNet | | |
|---|---|---|---|---|---|---|
| | Dataset | Images | Class | Dataset | Images | Class |
| ID data (Training) | CIFAR100 | 50,000 | 100 | ImageNet | 1.28M | 1000 |
| ID data (Testing) | CIFAR100 | 10,000 | 100 | ImageNet | 50,000 | 1000 |
| OOD data | CIFAR10 | 10,000 | 10 | ImageNet_O | 2,000 | / |
| | TinyImageNet | 7,498 | / | iNaturalist | 10,000 | 110 |
| | MNIST | 10,000 | 10 | OpenImage_O | 15,869 | / |
| | Texture | 5,640 | 47 | Species | 10,000 | 10 |
| | Places365 | 33,773 | / | MNIST | 10,000 | 10 |
| | SVHN | 26,032 | 10 | Texture | 5,640 | 47 |

### A.2    PERFORMANCE METRICS

This section provides more introduction to the performance measures used in our experiments. (1) FPR is the false positive rate of OOD examples when the true positive rate of ID examples is at 95%. It measures the portion of falsely recognized OOD when most of the ID samples are recalled. (2) AUC computes the area under the receiver operating characteristic curve of detecting OOD samples, evaluating the OOD detection performance. (3) AP measures the area under the precision-

recall curve, in which the OOD samples are treated as positive samples. (4) ACC calculates the classification accuracy of the ID data for the CIL models. Among all these metrics, only FPR95 is expected to have a lower value for a better model. Higher values indicate better performance for the other three metrics.

## B  BASELINE LIBRARY.

We include four different popular CIL models. iCaRL (Rebuffi et al., 2017) and WA (Zhao et al., 2020) are the regularization-based algorithms, BiC (Wu et al., 2019) is a replay-based algorithm, and FOSTER (Wang et al., 2022a) is a parameter-isolation-based method but it also uses data replay.

For OOD methods, there are two main categories, post-hoc-based and fine-tuning-based methods. For post-hoc-based methods, we include nine OOD detection methods: MSP (Hendrycks & Gimpel, 2017), ODIN (Liang et al., 2018), Energy (Liu et al., 2020), MaxLogit (Basart et al., 2022), and GEN (Liu et al., 2023a) use the statistic based on prediction output of each test samples without using any training ID information, while the other four methods: ReAct (Sun et al., 2021), KLM (Basart et al., 2022), Relation (Kim et al., 2023), and NNGuide (Park et al., 2023) need the posterior information of training ID samples to obtain the OOD score. Therefore, if they need a whole label space of training ID data, we feed $T_t^{train}$ to them, which is the combination of memory data and current task training data. For fine-tuning-based methods, we include five methods: LogitNorm (Wei et al., 2022) and T2FNorm (Regmi et al., 2023) are the regularization-based methods that apply the normalization to calibrate the loss function, AUGMIX (Hendrycks et al., 2020) and REGMIX (Pinto et al., 2023) are the augmentation-based methods that apply the data augmentation to ID samples for enhancing the ID data training, while VOS (Du et al., 2022) are synthesis-based methods that generate pseudo-OOD samples to assist the detector training. Notably, VOS focuses on outlier synthesis, but applying their original training method to fine-tune the extra classifier is difficult since it was originally designed without the final linear classifier. Thus, to have a fair comparison, we apply the same energy regularization as our BER for them, which replaces our synthesized pseudo-OOD samples $\bar{x}_t$ with their synthesized outlier samples.

## C  IMPLEMENTATION DETAILS.

For the CIL pre-training, we employ default hyperparameters of these CIL models as stated in their original papers. For OOD fine-tuning, following (Yang et al., 2022; Zhang et al., 2023), we use the common setting with SGD optimizer, using a learning rate of $0.1$, a momentum of $0.9$, a weight decay of $0.0005$, and adjusting the learning rate using a cosine annealing learning rate schedule. The batch size is fixed at $128$ for all experiments. We freeze the feature extractor and original classifier, fine-tune the extra classifier for $10$ epochs, and keep other hyperparameters the same as the original paper. For our proposed BER baseline, following (Liu et al., 2020), we set $\tau = 1$, $\alpha = 0.1$, $p_{in} = -5$ and $p_{out} = -27$ by default. $\lambda = 0.002$ is used throughout the experiments. All results are averaged over three independent runs using different random seeds. All experiments are performed using $8$ NVIDIA RTX 3090.

## D  MORE RESULTS W.R.T. INCREASING INCREMENTAL STEPS

In this section, we provide more results for the different baselines w.r.t. increasing incremental steps in our paper. Particularly, following the results of iCaRL (Rebuffi et al., 2017) with CIFAR100 (Krizhevsky et al., 2009) in our paper, we provide the results based on Bic (Wu et al., 2019) in Fig. 4, WA (Zhao et al., 2020) in Fig. 5, FOSTER (Wang et al., 2022a) in Fig. 6. Furthermore, we show the relationship between ACC and OOD detection performance on these three CIL models in Fig. 7. We also show the average performance of all four CIL models with more OOD detectors (GEN (Liu et al., 2023a), KLM (Basart et al., 2022), and VOS (Du et al., 2022)) in Fig. 8.

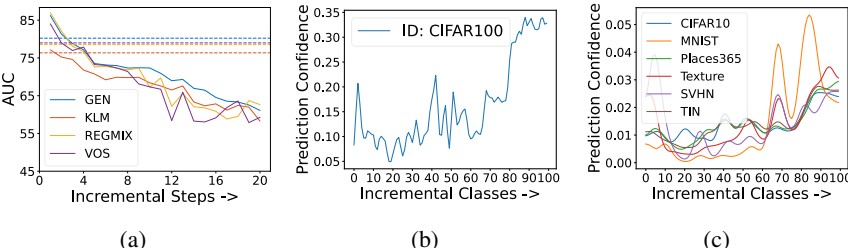

(a)                                        (b)                                        (c)

Figure 4: Qualitative results of the CIL model BiC with CIFAR100. **(a)** All four representative OOD detection methods experience a decreased AUC performance with increasing incremental steps, compared to themselves working on the full training data of all steps. **(b)** Mean prediction confidence of BiC on test samples from all incremental classes. **(c)** Mean prediction confidence of BiC classifying six OOD datasets into one of the ID classes based on the final incremental task.

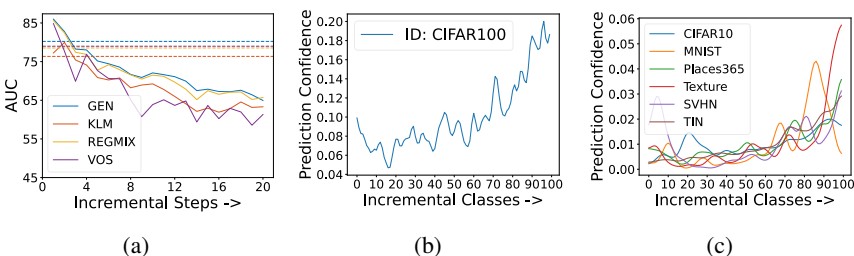

(a)                                        (b)                                        (c)

Figure 5: Qualitative results of the CIL model WA with CIFAR100. **(a)** All four representative OOD detection methods experience a decreased AUC performance with increasing incremental steps, compared to themselves working on the full training data of all steps. **(b)** Mean prediction confidence of WA on test samples from all incremental classes. **(c)** Mean prediction confidence of WA classifying six OOD datasets into one of the ID classes based on the final incremental task.

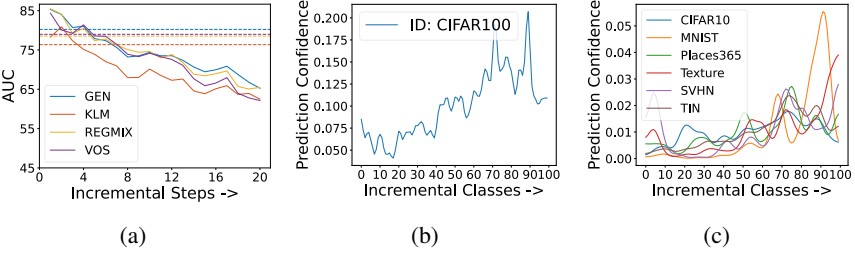

(a)                                        (b)                                        (c)

Figure 6: Qualitative results of the CIL model FOSTER with CIFAR100. **(a)** All four representative OOD detection methods experience a decreased AUC performance with increasing incremental steps, compared to themselves working on the full training data of all steps. **(b)** Mean prediction confidence of FOSTER on test samples from all incremental classes. **(c)** Mean prediction confidence of FOSTER classifying six OOD datasets into one of the ID classes based on the final incremental task.

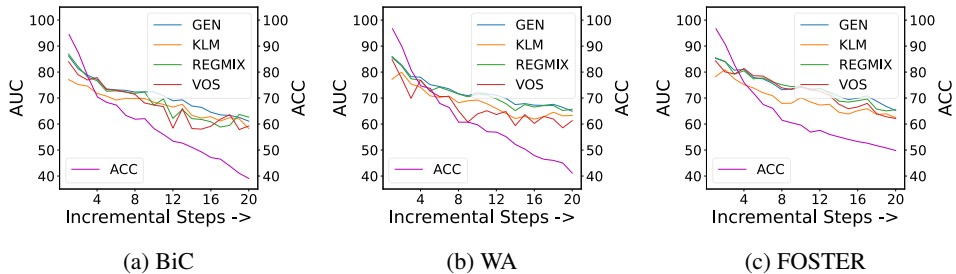

(a) BiC        (b) WA        (c) FOSTER

Figure 7: Average performance of four representative OOD methods on six OOD datasets at each incremental step, where the different CIL models are used. ACC is the accuracy of CIL models on CIFAR100 at each step.

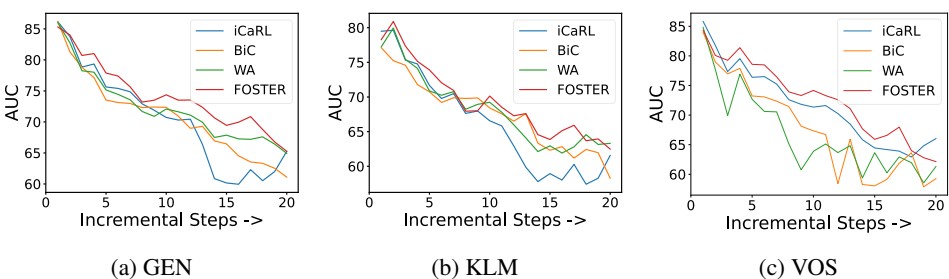

(a) GEN        (b) KLM        (c) VOS

Figure 8: Average performance of CIL models with different OOD detectors on six OOD datasets at each incremental step on CIFAR100.

# E  FINE-GRAINED EXPERIMENTAL RESULTS ON NEAR- AND FAR-OOD DETECTION DATASETS

Following the experiment results in our paper, we provide more fine-grained results (on near-OOD datasets and far-OOD datasets, respectively) at different step sizes in Tables 6, 7, and 8, in which we also add the AP results of the OOD detection performance.

Table 6: Detailed results for those in Table 1 on **near- and far-OOD detection datasets** at the **step size of** $k = 5$ **for CIFAR 100** and at the **step size of** $k = 50$ **for ImageNet1K**. The **best** and second-best performance per dataset in the fine-tuning-based methods are highlighted. The upper, middle, lower parts of the table are for AUC, AP, and FPR performance, respectively.

**ID Dataset: CIFAR100**

| AUC↑ | iCaRL Near | iCaRL Far | BiC Near | BiC Far | WA Near | WA Far | FOSTER Near | FOSTER Far | Average Near | Average Far |
|---|---|---|---|---|---|---|---|---|---|---|
| *Average CIL accuracy* | 58.20 | | 55.87 | | 61.44 | | 63.51 | | 59.76 | |
| *-Post-hoc-based OOD methods* | | | | | | | | | | |
| MSP | 67.21 | 66.73 | 67.69 | 69.15 | 70.22 | 68.68 | 70.14 | 69.94 | 68.81 | 68.62 |
| ODIN | 70.97 | 69.91 | 69.22 | 71.56 | 72.98 | 71.06 | 72.25 | 73.21 | 71.36 | 71.44 |
| Energy | 71.23 | 69.73 | 68.56 | 70.05 | 73.19 | 71.38 | 72.98 | 74.34 | 71.49 | 71.38 |
| MaxLogit | 71.16 | 69.66 | 68.85 | 70.38 | 73.16 | 71.26 | 72.98 | 74.25 | 71.54 | 71.39 |
| GEN | 71.18 | 69.99 | 69.44 | 71.61 | 73.30 | 71.68 | 73.34 | 74.65 | 71.81 | 71.98 |
| ReAct | 68.67 | 70.98 | 66.29 | 71.65 | 72.30 | 74.16 | 70.81 | 76.03 | 69.52 | 73.20 |
| KLM | 66.78 | 65.92 | 66.44 | 68.09 | 68.55 | 68.01 | 69.47 | 69.35 | 67.81 | 67.84 |
| Relation | 61.30 | 68.85 | 65.38 | 73.27 | 67.53 | 74.07 | 68.42 | 74.52 | 65.66 | 72.68 |
| NNGuide | 67.33 | 71.73 | 66.81 | 72.65 | 68.53 | 73.13 | 68.97 | 76.03 | 67.91 | 73.38 |
| Average | 68.43 | 69.28 | 67.63 | 70.93 | 71.08 | 71.49 | 71.04 | 73.59 | 69.55 | 71.32 |
| *-Fine-tuning-based OOD methods* | | | | | | | | | | |
| LogitNorm | 71.22 | 69.70 | 68.28 | 69.70 | 72.11 | 70.64 | 72.59 | 73.66 | 71.05 | 70.93 |
| T2FNorm | 71.37 | 69.99 | 68.03 | 70.37 | 71.45 | 70.62 | 72.50 | 73.64 | 70.84 | 71.16 |
| AUGMIX | 71.47 | 69.67 | 68.12 | 68.92 | 72.15 | 70.58 | 72.74 | 73.32 | 71.12 | 70.62 |
| REGMIX | 71.99 | 70.38 | 68.08 | 69.44 | 72.94 | 70.69 | 73.56 | 73.90 | 71.64 | 71.10 |
| VOS | 71.96 | 71.33 | 61.91 | 70.48 | 62.23 | 68.98 | 72.18 | 72.96 | 67.07 | 70.94 |
| BER (Ours) | 72.54 | 72.85 | 69.87 | 72.28 | 68.21 | 74.60 | 73.60 | 74.49 | 71.06 | 73.55 |
| Average | 71.60 | 70.21 | 67.36 | 69.78 | 70.34 | 70.30 | 72.71 | 73.50 | 70.34 | 70.95 |
| Average (All) | 69.56 | 69.61 | 67.36 | 70.52 | 70.76 | 71.06 | 71.64 | 73.56 | 69.83 | 71.19 |

**ID Dataset: ImageNet1K**

| AUC↑ | iCaRL Near | iCaRL Far | BiC Near | BiC Far | WA Near | WA Far | FOSTER Near | FOSTER Far | Average Near | Average Far |
|---|---|---|---|---|---|---|---|---|---|---|
| *Average CIL accuracy* | 40.86 | | 42.26 | | 45.99 | | 45.96 | | 43.77 | |
| MSP | 61.82 | 61.98 | 65.12 | 58.92 | 62.50 | 60.07 | 64.12 | 68.12 | 63.39 | 62.27 |
| ODIN | 64.80 | 68.40 | 68.11 | 66.65 | 63.29 | 63.83 | 66.19 | 75.10 | 65.60 | 68.50 |
| Energy | 62.62 | 63.07 | 67.58 | 63.37 | 62.00 | 63.50 | 64.59 | 73.29 | 64.20 | 65.81 |
| MaxLogit | 62.87 | 63.17 | 67.11 | 62.62 | 62.65 | 62.01 | 65.02 | 72.34 | 64.41 | 65.03 |
| GEN | 63.55 | 64.72 | 65.69 | 62.61 | 64.69 | 61.15 | 65.23 | 71.47 | 64.79 | 64.99 |
| ReAct | 57.92 | 51.17 | 63.42 | 55.09 | 62.11 | 59.59 | 59.26 | 61.31 | 60.68 | 56.79 |
| KLM | 63.55 | 63.46 | 63.62 | 61.55 | 63.40 | 62.74 | 65.73 | 67.15 | 64.06 | 63.73 |
| Relation | 58.73 | 69.95 | 66.19 | 66.80 | 63.22 | 62.87 | 61.05 | 68.37 | 62.30 | 67.00 |
| NNGuide | 61.20 | 66.61 | 70.63 | 68.36 | 58.74 | 70.33 | 63.94 | 78.02 | 63.63 | 70.83 |
| Average | 61.87 | 63.61 | 66.39 | 62.89 | 62.51 | 62.90 | 63.93 | 70.57 | 63.67 | 64.99 |
| LogitNorm | 61.49 | 63.09 | 66.52 | 65.86 | 61.25 | 62.25 | 63.53 | 70.53 | 63.20 | 65.43 |
| T2FNorm | 62.65 | 63.16 | 66.47 | 64.45 | 62.00 | 63.21 | 64.16 | 71.34 | 63.82 | 65.54 |
| AUGMIX | 62.79 | 60.99 | 67.43 | 63.05 | 61.96 | 64.72 | 65.97 | 71.09 | 64.54 | 64.96 |
| REGMIX | 66.05 | 59.70 | 68.13 | 62.45 | 64.26 | 60.36 | 67.13 | 69.27 | 66.39 | 62.94 |
| VOS | 62.17 | 59.65 | 67.03 | 63.16 | 61.48 | 63.28 | 65.24 | 69.71 | 63.98 | 63.95 |
| BER (Ours) | 64.04 | 62.29 | 68.96 | 65.25 | 64.00 | 64.30 | 67.16 | 73.70 | 66.04 | 66.38 |
| Average | 63.03 | 61.32 | 67.42 | 63.79 | 62.19 | 62.76 | 65.21 | 70.39 | 64.39 | 64.56 |
| Average (All) | 62.28 | 62.79 | 66.65 | 63.21 | 62.40 | 62.85 | 64.39 | 70.51 | 63.93 | 64.84 |

**AP — ID Dataset: CIFAR100**

| AP↑ | iCaRL Near | iCaRL Far | BiC Near | BiC Far | WA Near | WA Far | FOSTER Near | FOSTER Far | Average Near | Average Far |
|---|---|---|---|---|---|---|---|---|---|---|
| *Average CIL accuracy* | 58.20 | | 55.87 | | 61.44 | | 63.51 | | 59.76 | |
| *-Post-hoc-based OOD methods* | | | | | | | | | | |
| MSP | 62.73 | 77.05 | 62.57 | 76.91 | 65.46 | 77.77 | 65.08 | 78.70 | 63.96 | 77.61 |
| ODIN | 66.72 | 76.69 | 64.51 | 77.01 | 68.34 | 77.22 | 67.73 | 78.80 | 66.83 | 77.43 |
| Energy | 66.91 | 76.88 | 63.27 | 76.01 | 68.44 | 77.48 | 68.44 | 79.37 | 66.76 | 77.44 |
| MaxLogit | 66.79 | 76.94 | 63.73 | 76.48 | 68.34 | 77.69 | 68.38 | 79.46 | 66.81 | 77.64 |
| GEN | 66.82 | 77.17 | 64.69 | 77.14 | 68.47 | 77.76 | 68.63 | 79.70 | 67.15 | 77.94 |
| ReAct | 65.35 | 78.84 | 61.33 | 77.20 | 67.66 | 79.19 | 66.75 | 80.90 | 65.27 | 79.03 |
| KLM | 62.17 | 76.80 | 62.02 | 78.65 | 62.92 | 78.44 | 64.09 | 79.07 | 62.80 | 78.24 |
| Relation | 61.13 | 78.48 | 61.63 | 80.21 | 64.64 | 82.66 | 65.50 | 84.27 | 63.23 | 81.41 |
| NNGuide | 66.45 | 80.63 | 62.47 | 78.56 | 67.86 | 81.45 | 66.92 | 78.62 | 65.93 | 79.81 |
| Average | 65.01 | 77.72 | 62.91 | 77.57 | 66.90 | 78.85 | 66.84 | 79.88 | 65.41 | 78.50 |
| *Fine-tuning-based OOD methods* | | | | | | | | | | |
| LogitNorm | 66.92 | 76.86 | 63.39 | 76.09 | 67.54 | 77.16 | 68.07 | 79.06 | 66.48 | 77.29 |
| T2FNorm | 67.03 | 77.10 | 63.24 | 76.85 | 66.88 | 77.44 | 67.93 | 79.19 | 66.27 | 77.64 |
| AUGMIX | 67.12 | 76.84 | 63.17 | 75.63 | 67.52 | 77.13 | 68.22 | 78.88 | 66.51 | 77.12 |
| REGMIX | 67.65 | 77.26 | 63.17 | 75.92 | 68.31 | 77.23 | 69.22 | 79.32 | 67.09 | 77.43 |
| VOS | 67.50 | 77.56 | 58.66 | 77.45 | 59.23 | 77.19 | 67.59 | 78.69 | 63.24 | 77.72 |
| BER (Ours) | 67.81 | 78.63 | 64.43 | 79.57 | 69.56 | 81.00 | 68.89 | 79.69 | 67.67 | 79.72 |
| Average | 67.24 | 77.12 | 62.33 | 76.39 | 65.92 | 77.23 | 68.21 | 79.03 | 65.92 | 77.44 |
| Average (All) | 65.81 | 77.51 | 62.70 | 77.15 | 66.54 | 78.27 | 67.33 | 79.58 | 65.59 | 78.12 |

**AP — ID Dataset: ImageNet1K**

| AP↑ | iCaRL Near | iCaRL Far | BiC Near | BiC Far | WA Near | WA Far | FOSTER Near | FOSTER Far | Average Near | Average Far |
|---|---|---|---|---|---|---|---|---|---|---|
| *Average CIL accuracy* | 40.86 | | 42.26 | | 45.99 | | 45.96 | | 43.77 | |
| MSP | 22.90 | 20.01 | 24.27 | 16.79 | 22.30 | 18.64 | 23.44 | 23.64 | 23.23 | 19.77 |
| ODIN | 24.11 | 26.47 | 26.44 | 20.77 | 22.84 | 21.73 | 24.24 | 31.43 | 24.41 | 25.10 |
| Energy | 22.40 | 21.21 | 25.32 | 17.72 | 21.63 | 19.61 | 22.91 | 27.15 | 23.17 | 21.08 |
| MaxLogit | 22.70 | 21.16 | 25.32 | 17.72 | 22.10 | 19.25 | 23.25 | 26.18 | 23.34 | 21.08 |
| GEN | 23.10 | 21.71 | 23.77 | 18.57 | 24.54 | 18.68 | 23.54 | 24.96 | 23.74 | 20.98 |
| ReAct | 21.14 | 15.51 | 23.59 | 15.14 | 23.24 | 16.84 | 21.55 | 17.84 | 22.38 | 16.33 |
| KLM | 24.17 | 20.25 | 23.96 | 19.58 | 23.91 | 21.54 | 26.15 | 23.51 | 24.54 | 21.22 |
| Relation | 25.75 | 20.03 | 26.93 | 21.43 | 22.16 | 19.27 | 22.46 | 25.88 | 24.33 | 21.65 |
| NNGuide | 24.47 | 18.74 | 23.33 | 24.38 | 23.98 | 25.34 | 25.62 | 30.94 | 23.86 | 24.59 |
| Average | 23.41 | 20.57 | 24.60 | 19.03 | 22.97 | 20.10 | 23.68 | 25.73 | 23.66 | 21.36 |
| LogitNorm | 21.68 | 20.97 | 25.67 | 19.59 | 21.05 | 19.16 | 22.21 | 23.25 | 22.65 | 20.74 |
| T2FNorm | 22.44 | 21.16 | 24.99 | 18.61 | 21.71 | 19.77 | 22.82 | 24.41 | 22.99 | 20.99 |
| AUGMIX | 24.63 | 19.31 | 23.96 | 20.14 | 23.79 | 18.54 | 24.83 | 22.17 | 24.30 | 20.04 |
| REGMIX | 25.98 | 17.61 | 27.59 | 17.62 | 24.36 | 17.61 | 25.92 | 20.54 | 25.96 | 18.34 |
| VOS | 23.73 | 19.60 | 26.89 | 20.23 | 23.48 | 19.63 | 24.86 | 24.53 | 24.74 | 21.00 |
| BER (Ours) | 26.86 | 22.65 | 27.04 | 22.05 | 26.54 | 22.23 | 27.46 | 24.02 | 26.98 | 22.74 |
| Average | 23.69 | 19.73 | 25.82 | 19.24 | 22.88 | 18.94 | 24.13 | 22.98 | 24.13 | 20.22 |
| Average (All) | 23.51 | 20.27 | 25.04 | 19.11 | 22.94 | 19.69 | 23.84 | 24.75 | 23.83 | 20.95 |

**FPR — ID Dataset: CIFAR100**

| FPR↓ | iCaRL Near | iCaRL Far | BiC Near | BiC Far | WA Near | WA Far | FOSTER Near | FOSTER Far | Average Near | Average Far |
|---|---|---|---|---|---|---|---|---|---|---|
| *Average CIL accuracy* | 58.20 | | 55.87 | | 61.44 | | 63.51 | | 59.76 | |
| *-Post-hoc-based OOD methods* | | | | | | | | | | |
| MSP | 87.34 | 88.12 | 88.23 | 86.72 | 85.16 | 86.59 | 86.10 | 85.47 | 86.71 | 86.72 |
| ODIN | 83.67 | 77.57 | 85.70 | 74.96 | 82.91 | 78.69 | 83.14 | 71.65 | 83.86 | 75.72 |
| Energy | 83.78 | 79.86 | 87.19 | 80.57 | 83.40 | 82.10 | 82.72 | 73.14 | 84.27 | 78.92 |
| MaxLogit | 83.91 | 80.85 | 86.80 | 81.57 | 83.07 | 82.90 | 82.94 | 74.54 | 84.18 | 79.97 |
| GEN | 83.84 | 81.19 | 85.88 | 76.37 | 82.77 | 81.82 | 82.56 | 74.14 | 83.76 | 78.38 |
| ReAct | 83.83 | 79.97 | 88.95 | 82.85 | 83.64 | 81.41 | 83.63 | 73.64 | 85.01 | 79.47 |
| KLM | 87.94 | 89.37 | 88.13 | 86.00 | 87.78 | 87.66 | 87.06 | 86.00 | 87.73 | 87.26 |
| Relation | 85.39 | 74.39 | 88.48 | 77.88 | 84.44 | 73.56 | 84.10 | 71.57 | 85.60 | 74.35 |
| NNGuide | 85.66 | 75.41 | 87.97 | 75.47 | 86.10 | 75.44 | 87.47 | 70.20 | 86.80 | 74.13 |
| Average | 85.04 | 80.75 | 87.48 | 80.27 | 84.36 | 81.13 | 84.41 | 75.59 | 85.32 | 79.44 |
| *Fine-tuning-based OOD methods* | | | | | | | | | | |
| LogitNorm | 83.77 | 79.88 | 86.75 | 81.70 | 83.71 | 82.11 | 83.01 | 73.66 | 84.31 | 79.34 |
| T2FNorm | 83.69 | 80.39 | 86.75 | 81.55 | 83.97 | 82.91 | 82.95 | 74.56 | 84.34 | 79.85 |
| AUGMIX | 83.69 | 79.98 | 86.92 | 81.28 | 83.71 | 82.55 | 82.53 | 74.20 | 84.21 | 79.50 |
| REGMIX | 83.47 | 80.26 | 87.06 | 81.94 | 83.31 | 84.80 | 81.93 | 75.38 | 83.94 | 80.59 |
| VOS | 83.67 | 77.91 | 89.30 | 72.05 | 88.44 | 77.70 | 83.55 | 73.84 | 86.24 | 75.38 |
| BER (Ours) | 82.78 | 75.00 | 86.44 | 73.51 | 83.02 | 76.52 | 82.81 | 70.98 | 83.76 | 74.00 |
| Average | 83.66 | 79.68 | 87.36 | 79.70 | 84.63 | 82.01 | 82.79 | 74.33 | 84.61 | 78.93 |
| Average (All) | 84.55 | 80.37 | 87.44 | 80.07 | 84.46 | 81.44 | 83.83 | 75.14 | 85.07 | 79.26 |

**FPR — ID Dataset: ImageNet1K**

| FPR↓ | iCaRL Near | iCaRL Far | BiC Near | BiC Far | WA Near | WA Far | FOSTER Near | FOSTER Far | Average Near | Average Far |
|---|---|---|---|---|---|---|---|---|---|---|
| *Average CIL accuracy* | 40.86 | | 42.26 | | 45.99 | | 45.96 | | 43.77 | |
| MSP | 91.12 | 90.83 | 91.25 | 94.89 | 92.70 | 93.14 | 91.83 | 89.02 | 91.72 | 91.97 |
| ODIN | 90.89 | 84.91 | 89.34 | 92.93 | 91.66 | 89.91 | 90.92 | 80.90 | 90.70 | 87.16 |
| Energy | 92.65 | 90.98 | 90.53 | 95.36 | 92.95 | 93.25 | 92.54 | 87.43 | 92.17 | 91.76 |
| MaxLogit | 92.46 | 90.94 | 91.00 | 95.30 | 92.92 | 92.97 | 92.41 | 88.00 | 92.20 | 91.80 |
| GEN | 92.12 | 90.50 | 92.30 | 95.37 | 90.40 | 93.52 | 92.00 | 89.90 | 91.71 | 92.32 |
| ReAct | 92.98 | 94.57 | 91.42 | 95.62 | 92.18 | 95.77 | 93.90 | 94.98 | 92.62 | 95.23 |
| KLM | 89.88 | 90.20 | 89.77 | 89.84 | 90.59 | 87.38 | 88.30 | 87.25 | 89.64 | 88.67 |
| Relation | 91.78 | 86.00 | 92.85 | 95.19 | 95.52 | 95.19 | 92.56 | 87.38 | 93.18 | 90.94 |
| NNGuide | 90.00 | 88.40 | 84.14 | 89.16 | 90.30 | 90.30 | 89.94 | 85.73 | 88.59 | 88.40 |
| Average | 91.54 | 89.70 | 90.29 | 93.74 | 92.14 | 92.38 | 91.60 | 87.84 | 91.39 | 90.91 |
| LogitNorm | 93.23 | 91.73 | 89.61 | 94.38 | 93.92 | 93.98 | 93.39 | 93.03 | 92.54 | 93.28 |
| T2FNorm | 92.62 | 91.12 | 90.77 | 95.36 | 92.97 | 92.58 | 92.67 | 90.70 | 92.26 | 92.44 |
| AUGMIX | 90.16 | 92.26 | 89.24 | 90.98 | 90.73 | 88.90 | 90.02 | 88.31 | 90.04 | 90.11 |
| REGMIX | 89.16 | 94.54 | 88.16 | 94.54 | 90.30 | 94.34 | 89.97 | 94.73 | 89.50 | 94.52 |
| VOS | 89.43 | 90.36 | 89.65 | 91.36 | 90.74 | 88.46 | 89.66 | 91.10 | 89.87 | 90.32 |
| BER (Ours) | 88.42 | 91.78 | 87.48 | 90.87 | 88.28 | 90.88 | 87.29 | 90.58 | 87.87 | 91.03 |
| Average | 91.00 | 91.99 | 89.49 | 93.32 | 91.73 | 91.65 | 91.14 | 91.57 | 90.84 | 92.13 |
| Average (All) | 91.35 | 90.52 | 90.00 | 93.59 | 91.99 | 92.12 | 91.44 | 89.17 | 91.19 | 91.35 |

Table 7: Fine-grained results on **near- and far-OOD detection datasets** at the **step size of** $k = 10$ **for CIFAR 100** and at the **step size of** $k = 100$ **for ImageNet1K**. The **best** and second-best performance per dataset in the fine-tuning-based methods are highlighted. The upper, middle, lower parts of the table are for AUC, AP, and FPR performance, respectively.

**AUC↑ — ID Dataset: CIFAR100 / ID Dataset: ImageNet1K**

| | iCaRL Near | iCaRL Far | BiC Near | BiC Far | WA Near | WA Far | FOSTER Near | FOSTER Far | Avg Near | Avg Far | iCaRL Near | iCaRL Far | BiC Near | BiC Far | WA Near | WA Far | FOSTER Near | FOSTER Far | Avg Near | Avg Far |
|---|---|---|---|---|---|---|---|---|---|---|---|---|---|---|---|---|---|---|---|---|
| *-Average CIL accuracy* | 60.08 | | 61.68 | | 65.88 | | 66.01 | | 63.41 | | 44.44 | | 49.63 | | 52.22 | | 52.29 | | 49.65 | |
| *-Post-hoc-based OOD methods* | | | | | | | | | | | | | | | | | | | | |
| MSP | 67.99 | 67.66 | 71.12 | 64.45 | 71.82 | 69.83 | 71.24 | 71.92 | 70.54 | 68.47 | 63.57 | 64.36 | 67.76 | 66.81 | 66.64 | 64.66 | 66.59 | 69.75 | 66.14 | 66.40 |
| ODIN | 71.10 | 69.53 | 73.95 | 67.67 | 74.18 | 70.69 | 73.37 | 75.37 | 73.15 | 70.81 | 67.06 | 71.13 | 70.54 | 72.19 | 67.62 | 68.34 | 67.61 | 75.49 | 68.21 | 71.79 |
| Energy | 71.72 | 69.67 | 74.01 | 67.63 | 74.62 | 72.23 | 74.12 | 76.54 | 73.62 | 71.52 | 64.87 | 67.17 | 69.35 | 71.79 | 65.96 | 66.78 | 66.80 | 74.59 | 66.75 | 70.33 |
| MaxLogit | 71.65 | 69.69 | 74.04 | 67.29 | 74.62 | 72.16 | 74.23 | 76.45 | 73.64 | 71.40 | 65.13 | 67.22 | 69.50 | 70.81 | 66.80 | 66.64 | 67.36 | 73.94 | 67.20 | 69.65 |
| GEN | 71.82 | 70.39 | 74.12 | 67.75 | 74.73 | 72.58 | 74.51 | 76.92 | 73.80 | 71.91 | 65.95 | 68.78 | 69.43 | 69.66 | 69.07 | 67.52 | 67.92 | 73.40 | 68.09 | 69.84 |
| ReAct | 67.63 | 69.38 | 73.24 | 69.31 | 73.54 | 75.54 | 75.16 | 76.64 | 72.39 | 72.72 | 58.88 | 49.21 | 66.53 | 66.63 | 65.11 | 61.31 | 65.10 | 65.25 | 63.90 | 60.60 |
| KLM | 67.82 | 67.36 | 69.91 | 64.45 | 70.46 | 69.61 | 70.98 | 71.52 | 69.79 | 68.23 | 65.77 | 67.91 | 68.62 | 64.72 | 65.99 | 63.72 | 69.18 | 72.31 | 67.39 | 67.16 |
| Relation | 58.11 | 67.55 | 67.34 | 64.55 | 72.13 | 78.12 | 67.40 | 74.55 | 66.25 | 71.19 | 63.33 | 72.47 | 68.51 | 70.30 | 67.61 | 67.31 | 65.67 | 72.42 | 66.28 | 70.62 |
| NNGuide | 68.34 | 74.04 | 67.37 | 69.43 | 72.53 | 77.13 | 74.45 | 76.63 | 70.67 | 74.31 | 64.88 | 69.85 | 69.43 | 73.05 | 66.06 | 73.97 | 63.58 | 75.72 | 65.99 | 73.15 |
| Average | 68.46 | 69.47 | 71.68 | 66.95 | 73.18 | 73.10 | 72.83 | 75.17 | 71.54 | 71.17 | 64.38 | 66.46 | 68.85 | 69.55 | 66.76 | 66.81 | 66.65 | 72.54 | 66.66 | 68.84 |
| *-Fine-tuning-based OOD methods* | | | | | | | | | | | | | | | | | | | | |
| LogitNorm | 71.88 | 69.72 | 73.52 | 69.20 | 73.56 | 71.28 | 73.86 | 75.18 | 73.20 | 71.34 | 64.30 | 66.85 | 68.25 | 72.13 | 65.31 | 68.28 | 66.36 | 70.78 | 66.06 | 69.51 |
| T2FNorm | 71.90 | 70.03 | 72.63 | 71.30 | 72.93 | 71.50 | 73.75 | 75.09 | 72.80 | 71.98 | 65.19 | 67.91 | 68.18 | 72.23 | 66.08 | 70.00 | 66.10 | 72.36 | 66.39 | 70.62 |
| AUGMIX | 72.50 | 69.82 | 73.19 | 69.58 | 73.63 | 71.35 | 74.33 | 74.96 | 73.41 | 71.43 | 64.94 | 75.91 | 67.75 | 76.97 | 65.71 | 72.87 | 67.23 | 69.06 | 66.41 | 73.70 |
| REGMIX | 73.12 | 70.64 | 72.97 | 69.91 | 73.81 | 71.11 | 74.74 | 75.51 | 73.66 | 71.79 | 67.86 | 63.51 | 69.41 | 70.68 | 67.70 | 66.97 | 68.96 | 70.36 | 68.48 | 67.88 |
| VOS | 73.41 | 71.76 | 67.55 | 66.23 | 65.31 | 70.08 | 73.80 | 74.97 | 70.02 | 70.76 | 63.69 | 71.47 | 64.94 | 70.13 | 64.14 | 67.94 | 67.19 | 72.57 | 64.99 | 70.53 |
| BER (Ours) | 73.72 | 74.39 | 75.08 | 70.20 | 75.43 | 77.44 | 75.20 | 77.00 | 74.86 | 74.76 | 67.40 | 71.60 | 70.94 | 73.94 | 68.51 | 70.16 | 68.10 | 75.41 | 68.74 | 72.78 |
| Average | 72.56 | 70.39 | 71.97 | 69.24 | 71.85 | 71.06 | 74.10 | 75.14 | 72.62 | 71.44 | 65.20 | 69.13 | 67.71 | 72.43 | 65.79 | 69.21 | 67.17 | 71.03 | 66.47 | 70.45 |
| Average (All) | 69.92 | 69.80 | 71.78 | 67.77 | 72.71 | 72.37 | 73.28 | 75.16 | 71.93 | 71.27 | 64.67 | 67.41 | 68.44 | 70.58 | 66.41 | 67.67 | 66.84 | 72.00 | 66.59 | 69.42 |

**AP↑ — ID Dataset: CIFAR100 / ID Dataset: ImageNet1K**

| | iCaRL Near | iCaRL Far | BiC Near | BiC Far | WA Near | WA Far | FOSTER Near | FOSTER Far | Avg Near | Avg Far | iCaRL Near | iCaRL Far | BiC Near | BiC Far | WA Near | WA Far | FOSTER Near | FOSTER Far | Avg Near | Avg Far |
|---|---|---|---|---|---|---|---|---|---|---|---|---|---|---|---|---|---|---|---|---|
| *-Average CIL accuracy* | 60.08 | | 61.68 | | 65.88 | | 66.01 | | 63.41 | | 44.44 | | 49.63 | | 52.22 | | 52.29 | | 49.65 | |
| *-Post-hoc-based OOD methods* | | | | | | | | | | | | | | | | | | | | |
| MSP | 62.92 | 77.47 | 66.42 | 75.73 | 66.93 | 78.47 | 66.22 | 79.97 | 65.62 | 77.91 | 24.27 | 21.29 | 26.87 | 22.73 | 25.14 | 19.96 | 25.38 | 25.00 | 25.41 | 22.24 |
| ODIN | 66.69 | 76.57 | 69.50 | 75.62 | 69.80 | 77.20 | 68.69 | 80.05 | 68.67 | 77.36 | 26.30 | 27.44 | 28.84 | 26.58 | 26.27 | 23.63 | 26.08 | 30.71 | 26.87 | 27.09 |
| Energy | 67.16 | 76.79 | 69.45 | 75.66 | 70.12 | 78.12 | 69.40 | 80.78 | 69.03 | 77.84 | 23.79 | 22.11 | 26.69 | 25.28 | 24.12 | 21.18 | 24.37 | 27.24 | 25.31 | 23.63 |
| MaxLogit | 67.02 | 76.91 | 69.47 | 75.73 | 70.03 | 78.37 | 69.41 | 80.90 | 68.98 | 77.98 | 24.18 | 22.41 | 27.17 | 24.60 | 24.90 | 20.69 | 25.00 | 26.82 | 25.31 | 23.63 |
| GEN | 67.18 | 77.32 | 69.52 | 75.81 | 70.16 | 78.40 | 69.61 | 81.13 | 69.12 | 78.16 | 27.41 | 23.06 | 28.65 | 21.59 | 25.54 | 26.09 | 26.59 | 23.52 | | |
| ReAct | 64.32 | 78.22 | 68.84 | 77.00 | 69.24 | 80.52 | 67.72 | 82.49 | 67.53 | 79.56 | 21.25 | 13.04 | 25.82 | 20.62 | 25.94 | 18.00 | 25.85 | 20.11 | 24.72 | 17.94 |
| KLM | 62.71 | 78.06 | 64.73 | 77.12 | 64.53 | 79.28 | 65.79 | 80.35 | 64.44 | 78.70 | 26.60 | 24.14 | 28.25 | 21.19 | 25.73 | 20.66 | 30.06 | 27.00 | 27.66 | 23.25 |
| Relation | 58.70 | 79.28 | 65.41 | 78.51 | 68.32 | 80.80 | 65.20 | 80.88 | 64.41 | 79.87 | 27.17 | 30.24 | 28.16 | 25.52 | 28.12 | 24.41 | 24.89 | 27.99 | 27.09 | 27.04 |
| NNGuide | 66.33 | 78.44 | 69.36 | 78.26 | 70.19 | 80.44 | 68.55 | 79.32 | 68.61 | 79.11 | 28.61 | 28.19 | 28.16 | 24.66 | 27.96 | 30.08 | 27.99 | 27.88 | | |
| Average | 64.78 | 77.67 | 68.08 | 76.60 | 68.81 | 79.07 | 67.82 | 80.74 | 67.37 | 78.52 | 25.06 | 23.59 | 27.54 | 24.23 | 26.34 | 21.64 | 26.13 | 26.78 | 26.27 | 24.06 |
| *-Fine-tuning-based OOD methods* | | | | | | | | | | | | | | | | | | | | |
| LogitNorm | 67.34 | 76.81 | 69.06 | 76.32 | 69.11 | 77.66 | 69.05 | 80.11 | 68.64 | 77.72 | 23.42 | 21.69 | 26.54 | 29.55 | 23.43 | 21.51 | 24.01 | 23.52 | 24.35 | 24.07 |
| T2FNorm | 67.28 | 77.07 | 68.19 | 77.89 | 68.33 | 78.02 | 68.92 | 80.24 | 68.18 | 78.30 | 24.12 | 22.60 | 25.87 | 27.36 | 24.15 | 22.77 | 24.25 | 24.90 | 24.60 | 24.41 |
| AUGMIX | 67.89 | 76.84 | 68.75 | 76.51 | 69.13 | 77.69 | 69.52 | 79.90 | 68.82 | 77.74 | 26.98 | 33.75 | 26.75 | 34.07 | 25.41 | 26.99 | 25.46 | 27.22 | 26.15 | 30.51 |
| REGMIX | 68.56 | 77.26 | 68.57 | 76.82 | 69.36 | 77.65 | 69.12 | 80.13 | 68.90 | 77.97 | 27.42 | 18.63 | 27.65 | 22.81 | 26.48 | 19.82 | 27.03 | 21.45 | 27.14 | 20.68 |
| VOS | 67.64 | 77.73 | 67.63 | 75.52 | 61.74 | 77.84 | 69.02 | 80.09 | 66.51 | 77.80 | 21.51 | 22.55 | 25.32 | 27.85 | 24.27 | 23.24 | 24.12 | 26.19 | 23.80 | 24.96 |
| BER (Ours) | 68.85 | 79.73 | 70.81 | 78.81 | 69.84 | 80.53 | 69.64 | 81.39 | 69.78 | 80.12 | 28.78 | 30.85 | 28.62 | 27.81 | 28.38 | 25.27 | 28.45 | 30.63 | 28.56 | 28.64 |
| Average | 67.74 | 77.14 | 68.44 | 76.61 | 67.53 | 77.77 | 69.13 | 80.09 | 68.21 | 77.91 | 24.69 | 23.84 | 26.43 | 28.33 | 24.75 | 22.87 | 24.97 | 24.66 | 25.21 | 24.93 |
| Average (All) | 65.84 | 77.48 | 68.21 | 76.60 | 68.35 | 78.61 | 68.29 | 80.51 | 67.67 | 78.30 | 24.93 | 23.68 | 27.14 | 25.69 | 25.77 | 22.08 | 25.72 | 26.02 | 25.89 | 24.37 |

**FPR↓ — ID Dataset: CIFAR100 / ID Dataset: ImageNet1K**

| | iCaRL Near | iCaRL Far | BiC Near | BiC Far | WA Near | WA Far | FOSTER Near | FOSTER Far | Avg Near | Avg Far | iCaRL Near | iCaRL Far | BiC Near | BiC Far | WA Near | WA Far | FOSTER Near | FOSTER Far | Avg Near | Avg Far |
|---|---|---|---|---|---|---|---|---|---|---|---|---|---|---|---|---|---|---|---|---|
| *-Average CIL accuracy* | 60.08 | | 61.68 | | 65.88 | | 66.01 | | 63.41 | | 44.44 | | 49.63 | | 52.22 | | 52.29 | | 49.65 | |
| *-Post-hoc-based OOD methods* | | | | | | | | | | | | | | | | | | | | |
| MSP | 88.31 | 88.34 | 85.34 | 87.80 | 85.38 | 86.37 | 85.97 | 84.43 | 86.25 | 86.73 | 90.01 | 89.42 | 89.26 | 88.99 | 91.20 | 93.00 | 90.37 | 87.20 | 90.21 | 89.65 |
| ODIN | 84.48 | 80.66 | 82.06 | 79.75 | 81.78 | 79.37 | 82.86 | 72.09 | 82.80 | 77.97 | 87.74 | 84.20 | 86.88 | 86.91 | 89.16 | 88.91 | 89.57 | 82.31 | 88.83 | 85.58 |
| Energy | 83.99 | 83.61 | 82.41 | 81.82 | 81.75 | 80.95 | 82.58 | 74.57 | 82.68 | 80.24 | 92.13 | 91.38 | 90.36 | 87.98 | 91.81 | 93.03 | 91.78 | 87.91 | 91.52 | 90.07 |
| MaxLogit | 84.57 | 84.39 | 82.26 | 83.04 | 82.12 | 82.44 | 82.72 | 75.80 | 82.92 | 81.42 | 91.72 | 90.72 | 89.88 | 88.68 | 91.53 | 93.03 | 91.26 | 87.53 | 91.08 | 90.24 |
| GEN | 84.26 | 84.33 | 82.57 | 82.48 | 81.81 | 81.22 | 82.52 | 75.20 | 82.79 | 80.81 | 91.18 | 89.97 | 89.21 | 90.88 | 87.00 | 91.62 | 90.64 | 88.47 | 89.51 | 90.24 |
| ReAct | 85.04 | 84.12 | 82.62 | 82.59 | 81.22 | 81.62 | 83.73 | 75.17 | 83.40 | 80.88 | 93.09 | 97.23 | 89.67 | 90.76 | 90.02 | 92.61 | 90.45 | 92.61 | 90.81 | 93.22 |
| KLM | 88.24 | 87.89 | 86.78 | 88.09 | 88.03 | 87.46 | 86.18 | 85.76 | 87.31 | 87.30 | 88.79 | 87.27 | 86.30 | 88.93 | 89.25 | 90.26 | 88.33 | 88.69 | 88.17 | 88.79 |
| Relation | 90.91 | 88.90 | 84.06 | 79.24 | 90.86 | 87.12 | 84.22 | 70.82 | 87.51 | 81.52 | 91.68 | 86.41 | 89.85 | 89.19 | 94.14 | 95.47 | 89.79 | 87.38 | 91.37 | 89.61 |
| NNGuide | 84.23 | 77.32 | 82.26 | 77.06 | 82.06 | 72.97 | 83.57 | 70.27 | 83.03 | 74.41 | 91.40 | 86.84 | 87.44 | 87.11 | 89.02 | 88.42 | 89.17 | 86.63 | 89.26 | 87.25 |
| Average | 86.00 | 84.40 | 83.37 | 82.43 | 84.00 | 82.17 | 83.82 | 76.01 | 84.30 | 81.25 | 90.99 | 89.27 | 88.85 | 88.76 | 90.35 | 91.84 | 90.15 | 87.64 | 90.08 | 89.38 |
| *-Fine-tuning-based OOD methods* | | | | | | | | | | | | | | | | | | | | |
| LogitNorm | 83.94 | 83.70 | 82.26 | 82.56 | 82.37 | 80.81 | 82.88 | 75.98 | 82.86 | 80.76 | 92.56 | 92.39 | 89.75 | 82.28 | 92.84 | 93.78 | 92.58 | 92.02 | 91.93 | 90.12 |
| T2FNorm | 84.16 | 84.13 | 83.15 | 82.50 | 83.16 | 81.91 | 82.75 | 76.87 | 83.31 | 81.35 | 91.88 | 91.00 | 90.87 | 85.53 | 91.79 | 91.92 | 91.57 | 89.71 | 91.53 | 89.54 |
| AUGMIX | 83.61 | 84.07 | 82.48 | 83.17 | 82.33 | 81.14 | 82.47 | 76.70 | 82.72 | 81.27 | 89.26 | 85.42 | 89.94 | 83.66 | 89.29 | 89.55 | 89.42 | 86.75 | 89.48 | 86.34 |
| REGMIX | 83.62 | 84.98 | 82.70 | 83.49 | 82.20 | 83.32 | 82.25 | 77.89 | 82.69 | 82.42 | 88.88 | 94.31 | 88.86 | 91.31 | 89.25 | 93.57 | 89.35 | 93.62 | 89.09 | 93.20 |
| VOS | 83.34 | 82.12 | 85.81 | 75.64 | 84.71 | 74.86 | 82.75 | 75.18 | 84.15 | 76.95 | 92.36 | 90.29 | 91.35 | 85.71 | 90.14 | 89.48 | 90.24 | 85.72 | 91.02 | 87.80 |
| BER (Ours) | 83.61 | 76.49 | 85.30 | 73.99 | 80.94 | 72.89 | 82.17 | 70.78 | 83.00 | 73.54 | 88.31 | 85.91 | 85.32 | 89.75 | 87.18 | 88.40 | 88.45 | 85.39 | 87.31 | 87.36 |
| Average | 83.73 | 83.80 | 83.28 | 81.47 | 82.95 | 80.41 | 82.62 | 76.52 | 83.15 | 80.55 | 90.99 | 90.68 | 90.15 | 85.70 | 90.66 | 91.66 | 90.63 | 89.56 | 90.61 | 89.40 |
| Average (All) | 85.19 | 84.19 | 83.34 | 82.09 | 83.62 | 81.54 | 83.39 | 76.19 | 83.89 | 81.00 | 90.99 | 89.77 | 89.31 | 87.67 | 90.46 | 91.78 | 90.32 | 88.33 | 90.27 | 89.39 |

Table 8: Detailed results for those in Table 3 on **near- and far-OOD detection datasets** at the **step size of** $k = 20$ **for CIFAR 100** and at the **step size of** $k = 200$ **for ImageNet1K**. The **best** and second-best performance per dataset in the fine-tuning-based methods are highlighted. The upper, middle, lower parts of the table are for AUC, AP, and FPR performance, respectively.

| AUC↑ | iCaRL Near | iCaRL Far | BiC Near | BiC Far | WA Near | WA Far | FOSTER Near | FOSTER Far | Average Near | Average Far | iCaRL Near | iCaRL Far | BiC Near | BiC Far | WA Near | WA Far | FOSTER Near | FOSTER Far | Average Near | Average Far |
|---|---|---|---|---|---|---|---|---|---|---|---|---|---|---|---|---|---|---|---|---|
| | | | | | | | | ID Dataset: CIFAR100 | | | | | | | | | ID Dataset: ImageNet1K | | | |
| *Average CIL accuracy* | 62.65 | | 64.14 | | 68.05 | | 68.75 | | 65.90 | | 48.85 | | 53.30 | | 58.42 | | 57.84 | | 54.60 | |
| *-Post-hoc-based OOD methods* | | | | | | | | | | | | | | | | | | | | |
| MSP | 69.42 | 67.86 | 71.44 | 68.87 | 73.14 | 72.11 | 72.33 | 71.99 | 71.58 | 70.21 | 66.14 | 68.23 | 70.66 | 72.84 | 69.22 | 73.77 | 69.35 | 76.01 | 68.84 | 72.71 |
| ODIN | 72.67 | 70.68 | 74.14 | 70.11 | 75.06 | 73.87 | 74.02 | 74.67 | 73.97 | 72.33 | 69.56 | 74.95 | 72.70 | 80.40 | 69.88 | 75.29 | 70.83 | 79.00 | 70.74 | 77.41 |
| Energy | 73.62 | 71.75 | 74.19 | 70.13 | 75.32 | 75.17 | 75.64 | 76.39 | 74.69 | 73.36 | 67.58 | 73.41 | 71.44 | 80.20 | 68.36 | 74.95 | 69.92 | 78.17 | 69.33 | 76.68 |
| MaxLogit | 73.58 | 71.69 | 74.31 | 70.25 | 75.45 | 75.10 | 75.65 | 76.30 | 74.75 | 73.33 | 67.88 | 73.09 | 71.87 | 78.76 | 69.29 | 74.66 | 70.50 | 78.46 | 69.89 | 76.24 |
| GEN | 73.74 | 72.22 | 74.56 | 70.59 | 75.58 | 75.25 | 75.88 | 76.88 | 74.94 | 73.73 | 68.77 | 72.94 | 72.27 | 78.04 | 70.97 | 73.09 | 71.27 | 79.00 | 70.82 | 75.77 |
| ReAct | 70.33 | 72.22 | 73.64 | 73.72 | 74.16 | 78.15 | 74.81 | 78.60 | 73.23 | 75.67 | 60.87 | 56.69 | 68.14 | 72.73 | 63.74 | 61.87 | 66.82 | 60.83 | 64.89 | 63.03 |
| KLM | 69.38 | 68.96 | 71.08 | 69.75 | 72.76 | 72.33 | 72.62 | 72.40 | 71.46 | 70.86 | 68.67 | 71.19 | 71.37 | 68.92 | 69.94 | 68.47 | 72.37 | 75.89 | 70.59 | 71.23 |
| Relation | 60.26 | 67.92 | 67.36 | 69.97 | 72.90 | 77.89 | 66.61 | 73.61 | 66.78 | 72.35 | 69.03 | 75.66 | 72.58 | 80.21 | 69.52 | 75.30 | 69.76 | 80.29 | 70.22 | 77.87 |
| NNGuide | 71.30 | 75.56 | 71.33 | 71.20 | 73.40 | 77.72 | 72.02 | 78.30 | 72.01 | 75.69 | 68.64 | 76.97 | 75.00 | 83.84 | 68.53 | 78.64 | 70.03 | 79.75 | 70.55 | 79.80 |
| Average | 70.48 | 70.98 | 72.45 | 70.51 | 74.20 | 75.29 | 73.29 | 75.46 | 72.61 | 73.06 | 67.46 | 71.46 | 71.78 | 77.33 | 68.83 | 72.95 | 70.09 | 76.38 | 69.54 | 74.53 |
| *-Fine-tuning-based OOD methods* | | | | | | | | | | | | | | | | | | | | |
| LogitNorm | 74.05 | 71.59 | **73.34** | 70.23 | 74.81 | 74.03 | 75.63 | 75.25 | 74.46 | 72.78 | 68.55 | 71.11 | 70.12 | 79.30 | 67.75 | 71.92 | 70.00 | 74.85 | 69.11 | 74.29 |
| T2FNorm | 74.00 | 72.01 | 72.31 | 71.12 | 74.69 | 74.23 | 75.80 | 75.65 | 74.20 | 73.25 | 68.68 | 73.98 | 70.42 | 81.78 | 69.08 | 74.69 | 69.34 | 76.24 | 69.38 | 76.67 |
| AUGMIX | **74.65** | 71.43 | 73.06 | 69.52 | **75.30** | 73.87 | **76.22** | 75.20 | **74.81** | 72.50 | 67.42 | **79.12** | 71.80 | **86.56** | 67.22 | **78.09** | 68.86 | **80.72** | 68.83 | **81.12** |
| REGMIX | 74.54 | 71.84 | 72.97 | 70.94 | 75.19 | 74.57 | 76.00 | 75.53 | 74.67 | 73.22 | **69.64** | 69.41 | 70.05 | 78.26 | 69.44 | 73.16 | 70.85 | 73.83 | **70.00** | 73.67 |
| VOS | 74.16 | 72.31 | 70.53 | 67.89 | 69.55 | 73.55 | 75.17 | 75.01 | 72.35 | 72.19 | 60.12 | 68.59 | 59.60 | 73.36 | 60.57 | 67.45 | 61.50 | 73.81 | 60.45 | 70.80 |
| **BER (Ours)** | 74.03 | **74.81** | 70.73 | **72.49** | 74.36 | **77.78** | 75.07 | **77.24** | 73.55 | **75.58** | 69.39 | 76.87 | **73.15** | 83.08 | **70.61** | 74.72 | **70.89** | 80.07 | **71.01** | 78.69 |
| Average | 74.28 | 71.84 | 72.44 | 69.94 | 73.91 | 74.05 | 75.76 | 75.33 | 74.10 | 72.79 | 66.88 | 72.44 | 68.40 | 79.85 | 66.81 | 73.06 | 68.11 | 75.89 | 67.55 | 75.31 |
| Average (All) | 71.84 | 71.29 | 72.45 | 70.31 | 74.10 | 74.85 | 74.17 | 75.41 | 73.14 | 72.96 | 67.25 | 71.81 | 70.57 | 78.23 | 68.11 | 72.99 | 69.38 | 76.20 | 68.83 | 74.81 |

| AP↑ | iCaRL Near | iCaRL Far | BiC Near | BiC Far | WA Near | WA Far | FOSTER Near | FOSTER Far | Average Near | Average Far | iCaRL Near | iCaRL Far | BiC Near | BiC Far | WA Near | WA Far | FOSTER Near | FOSTER Far | Average Near | Average Far |
|---|---|---|---|---|---|---|---|---|---|---|---|---|---|---|---|---|---|---|---|---|
| *Average CIL accuracy* | 62.65 | | 64.14 | | 68.05 | | 68.75 | | 65.90 | | 48.85 | | 53.30 | | 58.42 | | 57.84 | | 54.60 | |
| *-Post-hoc-based OOD methods* | | | | | | | | | | | | | | | | | | | | |
| MSP | 64.69 | 77.88 | 66.09 | 77.36 | 68.06 | 79.42 | 67.19 | 79.81 | 66.51 | 78.62 | 26.39 | 25.10 | 29.95 | 27.19 | 27.46 | 28.99 | 28.24 | 33.60 | 28.01 | 28.72 |
| ODIN | 68.22 | 76.99 | 68.99 | 75.95 | 70.69 | 78.59 | 69.52 | 79.47 | 69.35 | 77.75 | 29.08 | 32.24 | 31.33 | 37.17 | 29.02 | 31.30 | 29.01 | 36.89 | 29.61 | 34.40 |
| Energy | 69.22 | 77.93 | 68.94 | 76.22 | 70.91 | 79.51 | 71.12 | 80.44 | 70.05 | 78.53 | 25.81 | 28.64 | 28.82 | 35.39 | 26.01 | 29.60 | 27.01 | 32.20 | 26.91 | 31.46 |
| MaxLogit | 69.11 | 78.06 | 68.98 | 76.48 | 70.94 | 79.75 | 71.03 | 80.58 | 70.02 | 78.72 | 26.38 | 28.39 | 29.68 | 32.27 | 27.10 | 29.57 | 27.87 | 33.98 | 27.76 | 31.05 |
| GEN | 69.27 | 78.40 | 69.25 | 76.53 | 71.05 | 79.60 | 71.25 | 80.91 | 70.20 | 78.86 | 27.02 | 27.81 | 30.20 | 31.68 | 30.52 | 27.10 | 28.67 | 36.42 | 29.10 | 30.75 |
| ReAct | 66.69 | 79.64 | 68.56 | 78.62 | 70.19 | 81.96 | 70.37 | 82.46 | 68.95 | 80.67 | 22.29 | 17.14 | 28.76 | 26.53 | 25.82 | 20.59 | 28.13 | 18.63 | 26.25 | 20.72 |
| KLM | 65.01 | 79.43 | 65.97 | 79.45 | 67.14 | 80.87 | 67.58 | 81.20 | 66.42 | 80.24 | 29.76 | 26.16 | 30.73 | 24.36 | 29.31 | 23.31 | 34.15 | 28.59 | 30.99 | 25.61 |
| Relation | 58.89 | 81.39 | 65.14 | 80.71 | 69.00 | 85.23 | 65.80 | 82.40 | 64.71 | 82.43 | 28.20 | 32.80 | 31.11 | 30.76 | 33.80 | 27.37 | 28.26 | 28.12 | 30.34 | 29.76 |
| NNGuide | 69.13 | 82.22 | 69.16 | 79.13 | 71.16 | 83.33 | 71.80 | 82.63 | 70.31 | 81.83 | 31.80 | 42.55 | 25.52 | 29.69 | 32.09 | 29.09 | 33.44 | 29.75 | 30.65 | 29.36 |
| Average | 66.69 | 79.10 | 67.90 | 77.83 | 69.90 | 80.92 | 69.52 | 81.10 | 68.50 | 79.74 | 27.39 | 27.46 | 29.57 | 30.56 | 29.01 | 27.44 | 29.42 | 30.91 | 28.85 | 29.09 |
| *Fine-tuning-based OOD methods* | | | | | | | | | | | | | | | | | | | | |
| LogitNorm | 69.65 | 77.80 | 68.22 | 76.29 | 70.29 | 78.78 | 71.03 | 79.71 | 69.80 | 78.14 | 26.76 | 26.25 | 27.62 | 37.77 | 25.39 | 27.70 | 27.39 | 27.84 | 26.79 | 29.89 |
| T2FNorm | 69.53 | 78.22 | 67.29 | 77.07 | 70.04 | 79.23 | 71.12 | 80.11 | 69.50 | 78.66 | 26.88 | 29.40 | 27.96 | **41.73** | 26.59 | 30.40 | 27.04 | 29.86 | 27.12 | 32.85 |
| AUGMIX | 70.16 | 77.73 | 67.84 | 76.12 | 70.68 | 78.69 | 71.57 | 79.62 | 70.06 | 78.04 | 29.41 | 31.62 | 28.77 | 28.31 | 27.78 | 27.66 | 29.20 | 29.28 | 28.79 | 29.22 |
| REGMIX | 70.27 | 78.14 | 68.12 | 76.98 | 71.06 | 79.33 | 71.58 | 79.97 | 70.26 | 79.97 | 28.06 | 29.67 | 25.48 | 29.05 | 28.67 | 24.29 | 28.31 | 25.53 | | |
| VOS | 69.66 | 78.12 | 66.03 | 75.72 | 65.00 | 79.31 | 70.44 | 79.58 | 67.78 | 78.18 | 22.86 | 26.04 | 21.60 | 31.07 | 22.16 | 26.43 | 23.24 | 29.48 | 22.46 | 28.25 |
| **BER (Ours)** | 70.63 | **81.41** | **68.86** | **79.98** | 71.41 | **83.73** | 72.86 | **82.60** | 70.94 | **81.93** | **30.26** | **31.87** | **30.71** | 36.50 | **29.82** | **33.39** | **30.66** | **32.12** | **30.36** | **33.47** |
| Average | 69.85 | 78.00 | 67.50 | 76.44 | 69.41 | 79.07 | 71.15 | 79.80 | 69.48 | 78.32 | 26.87 | 27.20 | 26.80 | 33.71 | 25.92 | 27.53 | 27.18 | 28.15 | 26.69 | 29.15 |
| Average (All) | 67.82 | 78.71 | 67.76 | 77.33 | 69.72 | 80.26 | 70.10 | 80.64 | 68.85 | 79.23 | 27.20 | 27.37 | 28.58 | 31.68 | 27.91 | 27.47 | 28.62 | 29.92 | 28.08 | 29.11 |

| FPR↓ | iCaRL Near | iCaRL Far | BiC Near | BiC Far | WA Near | WA Far | FOSTER Near | FOSTER Far | Average Near | Average Far | iCaRL Near | iCaRL Far | BiC Near | BiC Far | WA Near | WA Far | FOSTER Near | FOSTER Far | Average Near | Average Far |
|---|---|---|---|---|---|---|---|---|---|---|---|---|---|---|---|---|---|---|---|---|
| *Average CIL accuracy* | 62.65 | | 64.14 | | 68.05 | | 68.75 | | 65.90 | | 48.85 | | 53.30 | | 58.42 | | 57.84 | | 54.60 | |
| *-Post-hoc-based OOD methods* | | | | | | | | | | | | | | | | | | | | |
| MSP | 86.59 | 87.49 | 86.66 | 88.30 | 84.62 | 84.80 | 85.18 | 84.94 | 85.76 | 86.38 | 88.25 | 85.89 | 86.92 | 86.19 | 89.12 | 83.14 | 87.95 | 77.41 | 88.06 | 83.16 |
| ODIN | 83.94 | 79.84 | 84.45 | 78.70 | 82.02 | 76.10 | 82.65 | 73.77 | 83.26 | 77.10 | 89.56 | 78.86 | 86.75 | 80.31 | 86.75 | 80.31 | 87.37 | 74.06 | 86.48 | 77.31 |
| Energy | 82.74 | 81.62 | 84.56 | 81.39 | 82.06 | 77.47 | 81.72 | 76.95 | 82.77 | 79.36 | 90.62 | 83.67 | 88.72 | 77.34 | 90.36 | 82.46 | 90.03 | 81.20 | 89.93 | 81.17 |
| MaxLogit | 82.62 | 81.99 | 84.56 | 83.23 | 81.85 | 79.10 | 81.44 | 77.54 | 82.62 | 80.47 | 89.24 | 84.32 | 87.31 | 82.91 | 89.54 | 82.69 | 89.14 | 78.44 | 89.08 | 81.84 |
| GEN | 82.86 | 82.23 | 84.43 | 82.28 | 81.67 | 78.23 | 81.24 | 76.96 | 82.55 | 79.92 | 89.24 | 84.32 | 87.31 | 82.91 | 85.46 | 86.07 | 88.44 | 76.02 | 87.61 | 82.33 |
| ReAct | 84.18 | 82.40 | 84.02 | 82.30 | 81.71 | 76.98 | 81.86 | 77.35 | 82.89 | 79.76 | 92.37 | 92.98 | 86.31 | 85.15 | 88.68 | 87.47 | 87.89 | 91.61 | 88.81 | 89.30 |
| KLM | 86.34 | 85.12 | 86.48 | 86.89 | 86.24 | 85.97 | 84.75 | 84.21 | 85.95 | 85.55 | 84.68 | 85.60 | 85.24 | 86.67 | 86.42 | 89.64 | 80.57 | 85.16 | 84.23 | 86.77 |
| Relation | 87.14 | 76.55 | 84.52 | 76.94 | 83.35 | 74.51 | 83.91 | 72.84 | 84.73 | 75.21 | 89.50 | 82.30 | 86.36 | 77.31 | 93.46 | 91.33 | 88.11 | 73.72 | 89.36 | 81.16 |
| NNGuide | 82.70 | 74.23 | 84.31 | 76.17 | 81.69 | 69.83 | 85.19 | 71.60 | 83.47 | 72.96 | 91.91 | 80.27 | 91.24 | 77.11 | 91.99 | 76.72 | 87.52 | 73.19 | 90.66 | 76.82 |
| Average | 84.35 | 81.27 | 84.91 | 81.80 | 82.78 | 78.11 | 83.10 | 77.35 | 83.78 | 79.63 | 89.16 | 84.21 | 87.30 | 81.23 | 89.09 | 84.43 | 87.45 | 78.98 | 88.25 | 82.21 |
| *Fine-tuning-based OOD methods* | | | | | | | | | | | | | | | | | | | | |
| LogitNorm | 82.61 | 82.17 | **84.77** | 83.31 | 82.23 | 78.12 | 81.87 | 77.98 | 82.87 | 80.40 | 90.18 | 87.16 | 89.59 | 72.99 | 91.49 | 85.73 | 90.31 | 87.28 | 90.39 | 83.29 |
| T2FNorm | 82.63 | 82.28 | 85.28 | 83.05 | 82.63 | 79.67 | 81.59 | 78.60 | 83.03 | 80.90 | 89.78 | **82.85** | 89.25 | **68.00** | 90.11 | 81.85 | 89.61 | 83.92 | 89.69 | 79.16 |
| AUGMIX | 82.34 | 82.77 | 84.96 | 84.02 | 81.91 | 78.99 | 81.34 | 78.74 | 82.64 | 81.13 | 89.08 | 85.59 | 88.27 | 77.35 | 92.17 | **77.26** | 88.27 | 74.44 | 89.45 | **78.66** |
| REGMIX | **81.88** | 83.27 | 85.08 | 84.43 | 81.84 | 79.75 | **81.07** | 79.61 | **82.47** | 81.77 | **88.40** | 92.04 | 88.75 | 85.22 | 88.44 | 87.50 | 87.88 | 90.78 | **88.37** | 88.88 |
| VOS | 82.83 | 80.67 | 85.24 | 74.80 | 86.31 | 71.70 | 82.50 | 76.54 | 84.22 | 75.93 | 91.28 | 86.32 | 88.06 | 88.46 | 91.31 | 84.44 | 90.04 | 84.19 | 90.17 | 85.85 |
| **BER (Ours)** | 82.70 | **73.77** | 85.96 | **73.97** | 81.59 | **66.79** | 81.14 | **71.99** | 82.85 | **71.63** | **87.71** | 85.57 | **86.06** | 76.91 | **85.40** | 87.91 | **87.63** | **72.72** | **86.70** | 80.78 |
| Average | 82.46 | 82.23 | 85.07 | 81.92 | 82.98 | 77.65 | 81.67 | 78.29 | 83.05 | 80.03 | 89.74 | 86.79 | 88.78 | 78.40 | 90.70 | 83.36 | 89.22 | 84.12 | 89.61 | 83.17 |
| Average (All) | 83.67 | 81.61 | 84.97 | 81.84 | 82.85 | 77.95 | 82.59 | 77.69 | 83.52 | 79.77 | 89.37 | 85.13 | 87.83 | 80.22 | 89.66 | 84.05 | 88.08 | 80.82 | 88.74 | 82.55 |

## F  THE OPENCIL BENCHMARK FRAMEWORK AND THE BER ALGORITHM

The full framework of the OpenCIL Benchmark for post-hoc-based OOD methods are given in Algorithm 1, and that for the fine-tuning-based OOD methods is given in Algorithm 2 below. Furthermore, the algorithm of our proposed method BER is given in Algorithm 3.

---

**Algorithm 1** : Post-hoc-based OOD Detection Framework in the OpenCIL Benchmark

---

**Class Incremental Training**

**Data**: A data memory $M$; A sequence of $c$ tasks ID data $T = \{T_1, T_2, ..., T_c\}$, $T_t = (X_t^{train}, X_t^{test}, Y_t)$

1: **for** each task **do**
2:    Obtain $t$-th $(1 \leq t \leq c)$ task training ID data $T_t^{train} = (X_t^{train}, Y_t) \cup M$
3:    **for** each iteration **do**
4:      Sample a mini-batch of ID training data from $T_t^{train}$
5:      Perform different CIL algorithms
6:    **end for**
7:    Update data memory $M$    *//\*for replay-based CIL models only\**
8:    Save current CIL model $\theta_t(\cdot)$
9: **end for**

**Output**: A well-trained CIL model $\theta(\cdot) = \{\theta_1(\cdot), \theta_2(\cdot), ..., \theta_c(\cdot)\}$

---

**Inference**

**Input**: A well-trained CIL model $\theta(\cdot) = \{\theta_1(\cdot), \theta_2(\cdot), ..., \theta_c(\cdot)\}$

**Data**: A sequence of $c$ tasks OOD data $X^{ood} = \{X_1^{ood}, X_2^{ood}, ..., X_c^{ood}\}$; A sequence of $c$ tasks ID data $T = \{T_1, T_2, ..., T_c\}$, $T_t = (X_t^{train}, X_t^{test}, Y_t)$

1: **for** each task **do**
2:    Obtain $t$-th $(1 \leq t \leq c)$ task testing ID data $T_t^{test} = X_1^{test} \cup X_2^{test} \cup ... \cup X_t^{test}$
3:    Obtain $t$-th $(1 \leq t \leq c)$ task testing OOD data $X_t^{ood}$
4:    Perform ID classification on CIL model $\theta_t(\cdot)$ based on $T_t^{test}$
5:    Perform different post-hoc OOD detection methods on CIL model $\theta_t(\cdot)$ based on $T_t^{test} \cup X_t^{ood}$
6: **end for**

**Output**: Average incremental accuracy; Average OOD detection performance

---

---

**Algorithm 2** : Fine-tuning-based OOD Detection Framework in the OpenCIL Benchmark

---

**Class Incremental Training**

**Data**: A data memory $M$; A sequence of $c$ tasks ID data $T = \{T_1, T_2, ..., T_c\}$, $T_t = (X_t^{train}, X_t^{test}, Y_t)$

 1: **for** each task **do**
 2:    Obtain $t$-th $(1 \le t \le c)$ task training ID data $T_t^{train} = (X_t^{train}, Y_t) \cup M$
 3:    **for** each iteration **do**
 4:      Sample a mini-batch of ID training data from $T_t^{train}$
 5:      Perform different CIL algorithms
 6:    **end for**
 7:    Update data memory $M$    *//\*for replay-based CIL models only\**
 8:    Save current CIL model $\theta_t(\cdot)$
 9: **end for**

**Output**: A well-trained CIL model $\theta(\cdot) = \{\theta_1(\cdot), \theta_2(\cdot), ..., \theta_c(\cdot)\}$

---

**OOD Method - Fine-tuning**

**Input**: A well-trained CIL model $\theta(\cdot) = \{\theta_1(\cdot), \theta_2(\cdot), ..., \theta_c(\cdot)\}$

**Data**: A data memory $M$; A sequence of $c$ tasks ID data $T = \{T_1, T_2, ..., T_c\}$, $T_t = (X_t^{train}, X_t^{test}, Y_t)$

 1: **for** each task **do**
 2:    Obtain $t$-th $(1 \le t \le c)$ task training ID data $T_t^{train} = (X_t^{train}, Y_t) \cup M$
 3:    Obtain $t$-th CIL model $\theta_t(\cdot) = \{\phi_t(\cdot), h_t(\cdot)\}$ which is composed of a feature extractor $\phi_t(\cdot)$ and a original classifier $h_t(\cdot)$
 4:    Initialize an extra classifier $f_t(\cdot)$
 5:    **for** each iteration **do**
 6:      freeze the $\phi_t(\cdot)$ and $h_t(\cdot)$
 7:      Sample a mini-batch of ID training data from $T_t^{train}$
 8:      Perform different training-time OOD detection method to finetune this extra classifier $f_t(\cdot)$ only on top of $\phi_t(\cdot)$.
 9:    **end for**
10:    Update data memory $M$    *//\*for replay-based CIL models only\**
11:    Save current CIL model $\theta_t(\cdot)$ with extra classifier $f_t(\cdot)$
12: **end for**

**Output**: A well-trained CIL model (the same as **Input**) with a finetuned classifier $(\theta(\cdot), f(\cdot)) = \{(\theta_1(\cdot), f_1(\cdot)), (\theta_2(\cdot), f_2(\cdot)), ..., (\theta_c(\cdot), f_c(\cdot))\}$

---

**Inference**

**Input**: A finetuned CIL model $(\theta(\cdot), f(\cdot)) = \{(\theta_1(\cdot), f_1(\cdot)), (\theta_2(\cdot), f_2(\cdot)), ..., (\theta_c(\cdot), f_c(\cdot))\}$

**Data**: A sequence of $c$ tasks OOD data $X^{ood} = \{X_1^{ood}, X_2^{ood}, ..., X_c^{ood}\}$; A sequence of $c$ tasks ID data $T = \{T_1, T_2, ..., T_c\}$, $T_t = (X_t^{train}, X_t^{test}, Y_t)$

 1: **for** each task **do**
 2:    Obtain $t$-th $(1 \le t \le c)$ task testing ID data $T_t^{test} = X_1^{test} \cup X_2^{test} \cup ... \cup X_t^{test}$
 3:    Obtain $t$-th $(1 \le t \le c)$ task testing OOD data $X_t^{ood}$
 4:    Perform ID classification on original CIL model $\theta_t(\cdot) = \{\phi_t(\cdot), h_t(\cdot)\}$ based on $T_t^{test}$
 5:    Perform different post-hoc OOD scoring function on finetuned classifier $\{\phi_t(\cdot), f_t(\cdot)\}$ based on $T_t^{test} \cup X_t^{ood}$
 6: **end for**

**Output**: Average incremental accuracy; Average OOD detection performance

---

---

**Algorithm 3** : Bi-directional Energy Regularization (BER)

---

**Input**: A well-trained CIL model $\theta(\cdot) = \{\theta_1(\cdot), \theta_2(\cdot), ..., \theta_c(\cdot)\}$

**Data**: A data memory $M$; A sequence of $c$ tasks ID data $T = \{T_1, T_2, ..., T_c\}$, $T_t = (X_t^{train}, X_t^{test}, Y_t)$

1: **for** each task **do**
2:   Obtain $t$-th ($1 \leq t \leq c$) task training ID data $T_t^{train} = (X_t^{train}, Y_t) \cup M$
3:   Obtain $t$-th CIL model $\theta_t(\cdot) = \{\phi_t(\cdot), h_t(\cdot)\}$ which is composed of a feature extractor $\phi_t(\cdot)$ and a original classifier $h_t(\cdot)$
4:   Initialize an extra classifier $f_t(\cdot)$
5:   **for** each iteration **do**
6:     Freeze the $\phi_t(\cdot)$ and $h_t(\cdot)$
7:     Sample a mini-batch of current task ID training data $\left\{(x_t^i, y_t^i)\right\}_{i=1}^{B}$ from $(X_t^{train}, Y_t)$
8:     Separate the current task batch into two identical parts with equal size
9:     Conduct mixup on the second part based on Eq.1
10:    Apply energy regularization on these two parts based on Eq.2 with extra classifier $f_t(\cdot)$
11:    Sample a mini-batch of old task ID training data $\left\{(x_o^i, y_o^i)\right\}_{i=1}^{B}$ ($1 \leq o < t$) from $M$
12:    Conduct mixup between the old task and current task ID training data based on Eq.3
13:    Apply energy regularization on this mixed data based on Eq.4 with extra classifier $f_t(\cdot)$
14:   **end for**
15:   Update data memory $M$
16:   Save current CIL model $\theta_t(\cdot)$ with extra classifier $f_t(\cdot)$
17: **end for**

**Output**: A well-trained CIL model (the same as **Input**) with a finetuned classifier $(\theta(\cdot), f(\cdot)) = \{(\theta_1(\cdot), f_1(\cdot)), (\theta_2(\cdot), f_2(\cdot)), ..., (\theta_c(\cdot), f_c(\cdot))\}$

---

