# OpenReview forum: "Out-of-Distribution Detection in Class Incremental Learning"
_ICLR.cc/2025/Conference — ICLR 2025 Conference Withdrawn Submission_

### Official Review · Reviewer_aqhc · 2024-10-28

**Soundness:** 2
**Presentation:** 2
**Contribution:** 2
**Rating:** 5
**Confidence:** 4

**Summary:**

This paper addresses the need of detecting out-of-distribution instances by class incremental learners in the open world on top of learning new classes without forgetting old knowledge. The authors propose a OpenCIL benchmark, which utilizes two commonly used class-incremental learning datasets and six out-of-distribution datasets. As baseline of the benchmark, the authors combine existing class-incremental learning and out-of-distribution detection methods and evaluate them on the datasets. To address the issue of increasing bias towards new class and out-of-distribution samples observed in the baseline methods, the authors proposed a Bi-directional Energy Regularization loss to improve the out-of-distribution capability of the class-incremental learning model. Specifically, the new task and old task energy regularization terms are used to modulate the energy prediction of out-of-distribution samples such that they can be distinguished from new class and old class samples respectively. Experimental results on the OpenCIL benchmark demonstrate the effectiveness of the proposed approach.

**Strengths:**

1. The paper address an important and challenging problem, which is to perform out-of-distribution detection in the open-world setting and thus, improves the applicability of incremental learning models in real world applications.
2. The proposed regularization term can be easily integrated into existing class incremental learning models to improve their capability in out-of-distribution detection.
3. Quantitative results demonstrates significant and consistent performance improvement.

**Weaknesses:**

1. The out-of-distribution dataset in OpenCIL benchmark is static, that is, these samples or classes remain as out-of-distribution throughout the whole incremental learning process. It does not address the scenario in the open-world where the incremental learner learns the out-of-distribution samples detected. In this case, these samples should no longer be detected as out-of-distribution by the model in the next incremental phase.
2.  The main findings stated in Section 3.2 needs further clarification. There is no strong justification of why 'catastrophic forgetting is more persistent in CIL models than OOD detection methods' given that the authors are comparing the decreasing rate of two different metrics. It is also unclear how the authors justify that old class samples are detected as out-of-distribution samples. From the prediction confidence in Figure 1, it is difficult to understand how the authors claim that old class samples have lower prediction confidence than out-of-distribution samples.
3. There seems to be some conflict proposed regularization terms. In Eq (3), the model penalizes in-distribution samples that produce energy lower than $p_{in}$ while penalizing pseudo-OOD samples that produce energy higher than $p_{out}$. On the other hand, in Eq (5), the model is penalizing augmented in-distribution samples that produce energy higher than $p_{in}$.
4. Limited class-incremental learning methods are used as baselines, where the most recent method is from year 2022. For a more comprehensive evaluation, it would be beneficial to include some of the latest released methods to better demonstrate the robustness and generalizability of proposed method.
5. Novelty is limited as the proposed method is a simple combination of mixup augmentation with the energy-based out-of-distribution detection from previous work (Liu et al., 2020).

**Questions:**

1. How would the method perform when the out-of-distribution dataset is dynamic? In other words, would the model be updated so that newly seen classes are no longer considered as out-of-distribution once they have been learned?
2. Further clarifications could be provided in Figure 1. How can we compare the prediction confidence between in-distribution and out-of-distribution samples on old classes and new classes to justify that old class samples are being misclassified as out-of-distribution samples?
3. For fine-tuning based methods, the prediction of in-distribution classification and out-of-distribution detection is separated. What is the purpose of comparing the 'prediction confidence' between in-distribution and out-of-distribution samples in Figure 1 given that the out-of-distribution branch of the model determines whether a sample is out-of-distribution?
4. Could the authors provide further clarification on Eq (3) and Eq (5)? It seems to also differ from the loss proposed in the original work (Liu et al., 2020).

---

### Official Review · Reviewer_1WAi · 2024-10-29

**Soundness:** 3
**Presentation:** 3
**Contribution:** 2
**Rating:** 5
**Confidence:** 4

**Summary:**

1.The variables p_in and p_out in the formulas in Sections 4.1 and 4.2 are not explained clearly. 2.There is an issue with the formula numbering in Algorithm 3 in the Appendix, where Eq.2 is incorrectly labeled as Eq.1. 3.The paper lacks sufficient novelty, as the CIL methods used are outdated, with the most recent work being from 2022. More recent methods should be considered to better demonstrate the novelty of the paper.

**Strengths:**

Originality: This paper designs a new benchmark named OpenCIL to establish the performance of the combination between Out-of-Distribution and Class Incremental Learning with two frameworks. Further more this paper proposes Bi-directional Energy Regularization to mitigate the imbalance between old classes and new classes as well as the Out-of-Distribution data.Quality: The experimental is comprehensive, and the results show a slight improvement over other methods. Clarity: The paper clearly defines the problem it aims to address. Significance: The significance of this paper lies in its contribution to provide unified structure for future researchers exploring the integration of OOD detection and CIL.

**Weaknesses:**

1.Lack of Comprehensive References: The paper does not thoroughly cite recent studies in the field, particularly the latest methods in Class Incremental Learning (CIL). The comparison experiments use CIL methods with the latest from 2022, which may not reflect the most current advancements. More comprehensive references and comparisons could enhance the credibility and persuasiveness of the approach.
2.Unclear Equations: In Sections 4.1 and 4.2, the key parameters pin and pout are not explained, which may hinder readers’ understanding of the equations. Additionally, there is a labeling error in Algorithm 3 in the Appendix, where Eq. 2 is mistakenly labeled as Eq. 1. Correcting these will improve the paper's professionalism and readability.
3.Lack of Advanced Comparison Methods: The comparison experiments do not include the most recent CIL methods, which may affect the reliability of the results. It is recommended to incorporate more recent CIL methods in the comparisons to provide a more thorough evaluation of the proposed framework.

**Questions:**

1. Could you supplement the references with more recent literature, especially CIL methods from 2023 onward, to better demonstrate the relative advantages of the proposed method?

2. In Sections 4.1 and 4.2, what are the specific meanings of pin and pout? What role do these parameters play in energy regularization? A detailed explanation of these parameters would help readers understand the equations.

3. There is an error in the equation labeling in Algorithm 3 in the Appendix. Could you review and correct the labels to ensure the document's professionalism?

4. Considering the extensibility of the proposed framework and experimental setup, have you considered including more advanced CIL methods as benchmarks to enhance the representativeness and rigor of the experiments?

---

### Official Review · Reviewer_k73f · 2024-11-03

**Soundness:** 3
**Presentation:** 2
**Contribution:** 2
**Rating:** 5
**Confidence:** 4

**Summary:**

This paper addresses a critical challenge in CIL: training models to learn new classes while retaining knowledge of previous ones and effectively detecting OOD samples, essential for open-world applications like autonomous systems. To evaluate OOD detection methods in CIL, the authors introduce OpenCIL, a benchmark featuring both post-hoc and fine-tuning-based methods across various CIL models and datasets. They also propose BE to mitigate biases toward new classes and OOD samples, using NTER and OTER to enhance OOD detection by synthesizing pseudo-OOD and augmented samples.

**Strengths:**

- The paper addresses the gap in **OOD detection** for **class incremental learning**, an area that has seen limited exploration. This contribution is novel, particularly in how it systematically enables a unified protocol for combining CIL models with diverse OOD detectors, which is essential for real-world applications.
- The explanations of the proposed BER method are clear, especially the roles of **New Task Energy Regularization (NTER)** and **Old Task Energy Regularization (OTER)**. These components are effectively tied back to the issues of bias and catastrophic forgetting, making the contributions and their impact easy to grasp.
- The paper is generally well-organized, with a clear structure that systematically introduces the problem, the proposed OpenCIL benchmark, and BER. The visual aids (e.g., Figs 1 and 2) are helpful in understanding the mechanisms and biases associated with CIL models in OOD contexts.

**Weaknesses:**

+ The paper briefly mentions hyperparameters like $\tau$ and $\alpha$ for the BER approach, but there’s no thorough exploration of how sensitive BER is to these parameters. A parameter sensitivity analysis could offer insights into how to optimize BER for different datasets and models, making it more user-friendly.
+ the complexity of the approach might pose challenges for real-world implementation, especially regarding **computational overhead** due to the additional synthesis and regularization components in NTER and OTER.
+ There are missing citations in the manuscript. For example, the paper introduces generative models, but generative-based methods [1, 2, 3] are missed without corresponding details in the bibliography.

[1]: Kong, Shu, and Deva Ramanan. "Opengan: Open-set recognition via open data generation." In ICCV. 2021.

[2]: Wang, Qizhou, et al. "Out-of-distribution detection with implicit outlier transformation." In ICLR, 2023.

[3]: Zheng, Haotian, et al. "Out-of-distribution Detection Learning with Unreliable Out-of-distribution Sources." In NeurIPS, 2023.

**Questions:**

+ Minor Comments
  - **Page 4, Figure 2 caption**: “principled frameworks for incorporating OOD detection methods” could be clearer if it stated explicitly that these frameworks apply to both post-hoc and fine-tuning-based methods.
  - **Equation (3)**: Consider defining variables like $p_{in}$ and $p_{out}$ more explicitly right after the equation to improve readability.
+ It would be valuable to see a breakdown of the computational trade-offs involved in implementing both NTER and OTER. Are there scenarios where only one of the two might be sufficient or more efficient?
+ Could the OOD detection capability of BER be further enhanced by augmenting the memory with synthesized samples from OOD data?

---

### Official Review · Reviewer_nP3v · 2024-11-04

**Soundness:** 2
**Presentation:** 2
**Contribution:** 2
**Rating:** 3
**Confidence:** 4

**Summary:**

This paper introduces OpenCIL, a benchmark for evaluating out-of-distribution (OOD) detection in class incremental learning (CIL), which is vital for practical deployment in open-world scenarios. The study identifies biases in CIL models towards OOD samples and new classes and proposes Bi-directional Energy Regularization (BER) to address these issues. BER applies energy regularization to both old and new classes, improving OOD detection. Experiments show that BER achieves effective performance improvements on the OpenCIL benchmark, enhancing CIL models' robustness.

**Strengths:**

1. The mixup-based pseudo OOD and old task data generation is interesting.
2. The experiment is extensive and comprehensive including most state-of-the-art OOD and CIL methods.

**Weaknesses:**

1. The motivation for benchmarking OOD in CIL is not strong, as OOD detection becomes more and more ineffective during CIL is intuitively predictable. There is no need to spend a lot of resources to illustrate a predictable finding.
2. Although the mixup-based pseudo-OOD samples and synthetic old task data are interesting, the OOD detection of the proposed BER still relies on existing energy-based methods. In terms of technical novelty, it is more appealing to propose a new OOD scoring mechanism dedicated to CIL.
3. For the OOD classifier, how to initialize it is unclear. Is it initialized from the ID classification classifier or inherited from the latest old OOD classifier?
4. I would disagree with the finding of lines 337-339. In most cases, the performance of fine-tuning-based OOD and post-hoc-based OOD methods are comparable. Besides, I am not sure why BER is only compared with fine-tuning-based OOD approaches. There are some cases in which BER cannot beat all post-hoc-based OOD methods. If fine-tuning-based BER cannot far exceed post-hoc-based OOD methods, I cannot figure out a reason to adopt BER, which even consumes much higher computation costs.
5. The paper presentation and organization have a lot to improve. For example, in Figure 3, there is no legend for dashed curves in subfigure (a); the legend in subfigure (b) denotes ACC performance and different OOD methods, which is confusing. Besides, the curves are the same type; The same curve with the same color denotes different methods among the three subfigures, which is hard to distinguish and read. In addition, the last finding (starts at line 337 on page 7) refers to Tables 1,2,3 that are on pages 9 and 10, which largely hurts the readability.

**Questions:**

See the weaknesses above.

---

### Note · Authors · 2024-11-25

I have read and agree with the venue's withdrawal policy on behalf of myself and my co-authors.